# PROGRA: PROGRESS-AWARE REINFORCEMENT LEARNING FOR MULTI-TURN FUNCTION CALLING

## ABSTRACT

Large language models (LLMs) have achieved impressive success in single-turn function calling, yet real-world applications such as travel planning or multi-stage data analysis typically unfold across multi-turn conversations. In these settings, LLMs must not only issue accurate function calls at each step but also maintain progress awareness, the ability to summarize past interactions and plan future actions to ensure coherent, long-horizon task execution. Existing approaches, however, either reduce multi-turn training to isolated single-turn samples, which neglects task-level planning, or employ end-to-end reinforcement learning (RL) that struggles with redundancy and lacks explicit integration of progress awareness. To overcome these limitations, we introduce PROGRA, a framework that explicitly incorporates progress awareness into LLM training for multi-turn function calling. PROGRA combines (i) a Progress Awareness Generation (PAG) pipeline, which automatically constructs datasets coupling conversation summaries with future task planning, and (ii) a Progress Awareness-Guided Reinforcement Learning (PAG-RL) algorithm, which integrates progress awareness into RL training to reduce contextual redundancy and improve alignment between local actions and global task completion. Empirical results on two public benchmarks demonstrate that PROGRA significantly outperforms existing methods, highlighting the effectiveness of progress awareness in enabling robust and efficient multi-turn function calling. Our code is made available at `https://anonymous.4open.science/r/Progra_ICLR26-57F0`.

## 1 INTRODUCTION

Large language models (LLMs) have shown remarkable progress in tool use, where function calls to external APIs extend their reasoning and execution capabilities (Feng et al., 2025; Acikgoz et al., 2025; Liu et al., 2024). Despite advances in enabling reliable function calling in *single-turn conversations*, real-world scenarios rarely conform to isolated interactions (Alkhouli et al., 2025; Lu et al., 2025). Instead, applications such as travel planning or enterprise workflows require *multi-turn conversations*, where the outcome of each turn directly influences subsequent turns, shaping not only the immediate response but also the entire conversation trajectory (Liu et al., 2025). This interdependence across turns places higher demands on LLMs: they must not only produce accurate responses or function calls at the current step but also maintain a precise, holistic progress awareness for task execution throughout the conversation (Sanders et al., 2022; Yin et al., 2025). Specifically, progress awareness encompasses both an accurate **summary** of the current interaction history, helping reduce redundancy in long contexts and assisting LLMs in decision-making, as well as **planning** for future task execution, allowing them to address the given task systematically(Rastogi et al., 2020). Limited capacity for progress awareness will cause LLMs to struggle with effectively managing long-horizon dependencies in conversations, leading to behaviours like repeatedly invoking functions or omitting parameters, becoming a bottleneck restricting improvements in multi-turn function calling.

Despite the significant impact of progress awareness, it is still overlooked in existing work on enhancing multi-turn function calling. Existing approaches to enhancing multi-turn function calling mainly rely on fine-tuning LLM with carefully curated dataset (Prabhakar et al., 2025b; Yin et al., 2025). In practice, these approaches often reconstruct multi-turn conversation datasets into samples consisting solely of single-turn function calls, training the model to improve its single-turn accuracy. This paradigm not only lacks the diversity and dynamism of real-world scenarios but also, by de-

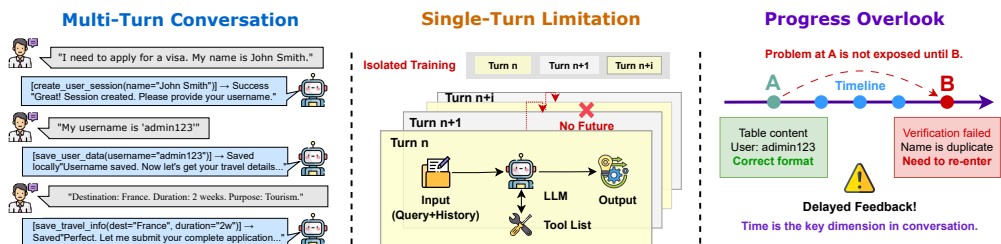

Figure 1: Limitations of single-turn training reside in overlooking progress in conversations

grading multi-turn task execution to single-turn question-answer predictions, neglects the awareness of the overall task progress during training (Sanders et al., 2022). Each single-turn sample completely excludes future conversations when training, preventing the effective training of the LLM's task planning capabilities. Moreover, a focus on single-turn accuracy leads the LLMs to overlook historical information that may not be immediately relevant to the current turn but could be referenced in future conversations, resulting in a bottleneck in training the model's ability to summarize history effectively.

Another direction is to leverage end-to-end reinforcement learning to optimize long-horizon returns in multi-turn settings (Singh et al., 2025; Chen et al., 2025b; Zhang et al., 2025). Recent RL-based approaches have improved the performance of function callings by modeling them as sequential decision-making. However, the end-to-end RL paradigm remains inefficient: as conversations grow, the input becomes increasingly redundant, exacerbating decision-making and optimization. Moreover, existing methods still lack explicit integration of progress awareness in training, preventing them from effectively aligning local actions with global task completion (Wang et al., 2025a).

To address the challenges of overlooking progress awareness in multi-turn conversations, we propose PROGRA, which explicitly incorporates progress awareness into model training to enhance its multi-turn function calling capabilities. Specifically, PROGRA consists of a **P**rogress **A**wareness **G**eneration pipeline (**PAG**) and a **P**rogress **A**wareness **G**uided **R**einforcement **L**earning training approach (**PAG-RL**). PAG automatically constructs a high-quality dataset that combines conversation history summaries with future task planning through a synthetic pipeline, thereby strengthening the ability of LLMs to generate progress awareness. Subsequently, PAG-RL explicitly applies the refined progress awareness to guide end-to-end reinforcement learning, reducing contextual redundancy during training and providing more efficient guidance for decision-making in dynamic real-world environments. Overall, our contributions focus on three key aspects:

1. **Progress Awareness Training.** To the best of our knowledge, PROGRA is the first work that identifies, formulates, and explicitly incorporates enhanced task progress awareness into the training of LLMs for multi-turn function calling.

2. **Progress Awareness Generation Pipeline.** We designed a novel progress awareness generation pipeline for automatically providing high-quality datasets to improve the capability of LLMs in progress awareness.

3. **Progress Awareness Guided RL.** We designed a progress awareness guided reinforcement learning algorithm, which enhances the model's training performance by explicitly incorporating progress awareness into end-to-end RL training, outperforming existing improvement strategies on two public benchmarks.

## 2 RELATED WORK

### 2.1 FUNCTION CALLING

Recent studies on the function-calling capabilities of large language models have increasingly transitioned from focusing on *single-turn* invocations (Liu et al., 2024) to exploring *multi-turn* scenarios (Chen et al., 2025a). While ToolLLM (Qin et al., 2023) constructs a large-scale dataset of massive real-world APIs. APIGen-MT (Prabhakar et al., 2025a) further develops a two-phase agentic pipeline that synthesizes verifiable multi-turn trajectories from blueprint tasks. These methods

address the *data bottleneck* for training, yet they remain essentially *data-driven*, lacking explicit modeling of global task progress. Another direction of enhancing the function calling ability is at prompt-engineering level, including reasoning–acting interleaving (Yao et al., 2023b), structured branching (Yao et al., 2023a), and self-reflection (Shinn et al., 2023) improve local robustness and error correction. However, these approaches emphasize local reasoning persistence rather than an explicit, evolving *progress progress* that connects intermediate calls with final task completion. In contrast, PROGRA aligns function call accuracy with overall task execution by explicitly incorporating progress awareness into model training.

## 2.2 MULTI-TURN REINFORCEMENT LEARNING

With the rapid development of reinforcement learning, several works introduce RL for multi-turn interaction. ARTIST (Singh et al., 2025) integrates outcome-based RL with dynamic tool routing, while RLFactory (Chai et al., 2025) offers a modular post-training pipeline for multi-turn orchestration. More recently, turn-level credit assignment (Zeng et al., 2025) and conversation-level preference optimization (Shi et al., 2024) explicitly frame multi-turn tool use as a sequential decision process, addressing delayed reward signals. On the algorithmic side, Group Relative Policy Optimization (GRPO) (Shao et al., 2024a) eliminates the critic through within-group normalization, enabling more stable and efficient updates. Many recent multi-turn RL studies have adopted GRPO to stabilize training in sparse-reward regimes (Mroueh et al., 2025). Despite these advances, existing RL agents rarely maintain an explicit *task progress awareness*. Without such awareness, agents often repeat calls, or omit critical steps in long-horizon workflows. Our approach differs by introducing *progress aware guidance* into the multi-turn RL, thereby reducing context redundancy and aligning local action choices with global task execution.

## 3 METHODOLOGY

In this section, we will provide a detailed introduction to PROGRA, which consists of two phases. In the first phase, PROGRA synthesizes a high-quality awareness dataset through a **P**rogress **A**wareness **G**eneration pipeline, namely **PAG**, and warmming up the trainable model and preliminarily enhance its progress awareness capabilities. The second phase applies a **P**rogress **A**wareness **G**uided **RL** algorithm, namely **PAG-RL**, using progress awareness during the rollout to guide decision-making and improving the overall performance.

### 3.1 PROBLEM FORMULATION AND NOTATION

We cast multi-turn function calling as a Markov Decision Process (MDP):

$$\mathcal{M} = (\mathcal{S}, \mathcal{A}, P_E, r, \gamma, \rho_0, H),$$

where $\mathcal{S}$ is the space of conversation prefixes (states), $\mathcal{A}$ is the textual action space (function invocations or user-facing messages), $P_E$ is the transition kernel in environment $E$, $r$ is the step reward, $\gamma \in (0,1]$ is the discount factor, $\rho_0$ is the initial state distribution, and $H$ is the horizon. A trainable LLM $\pi_\theta$ acts as the stochastic policy.

We index the conversation by turns $i \in \{1, \ldots, K\}$ (delimited by user queries $Q_i$) and intra-turn steps $j \in \{1, \ldots, T_i\}$. The state at $(i,j)$ is the entire dialogue prefix

$$S_{i,j} = \left\{ \left(Q_k, \{(A_{k,\ell}, O_{k,\ell})\}_{\ell=1}^{T_k}, A_k^{\text{msg}}\right)\right\}_{k=1}^{i-1} \cup \left(Q_i, \{(A_{i,\ell}, O_{i,\ell})\}_{\ell=1}^{j-1}\right), \tag{1}$$

where $A_{i,j} \in \mathcal{A}$ is either a structured function call with arguments or a terminal user-facing message $A_i^{\text{msg}}$ that ends turn $i$. $O_{i,j}$ is the observation returned by $E$ after applying action $A_{i,j}$. In step $j$, the policy will sample an action and receives feedbacks from the $E$:

$$A_{i,j} \sim \pi_\theta(\cdot \mid S_{i,j}), \quad (O_{i,j}, r_{i,j}) \sim P_E(\cdot \mid S_{i,j}, A_{i,j}),$$

and the conversation appends $(A_{i,j}, O_{i,j})$ to form $S_{i,j+1}$. A trajectory $\tau$ concatenates turns until solving $K$ user queries (or reach $H$ steps incompletely):

$$\tau = \left\{ \left(Q_i, \{(A_{i,j}, O_{i,j}, r_{i,j})\}_{j=1}^{T_i}, A_i^{\text{msg}}\right)\right\}_{i=1}^{K}, \quad R(\tau) = \sum_{i=1}^{K} \sum_{j=1}^{T_i} \gamma^{t(i,j)} r_{i,j}, \tag{2}$$

where $t(i,j)$ is the global step index. The learning objective is to maximize $\mathbb{E}_{\tau \sim \pi_\theta}[R(\tau)]$.

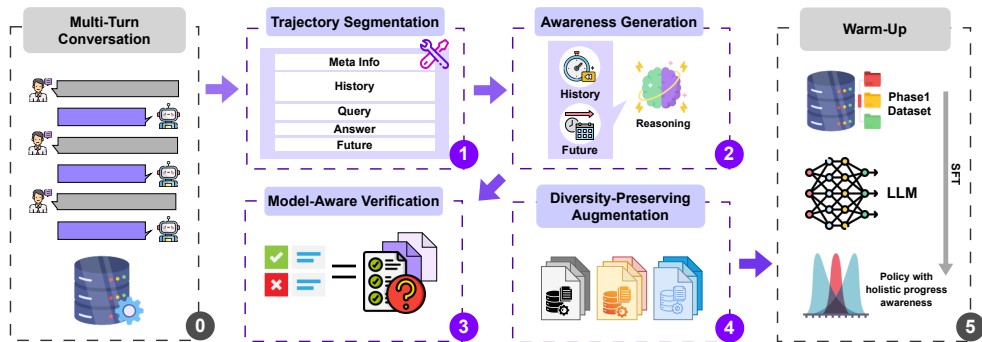

Figure 2: Overview of PAG. By summarizing history and planning future in Step 2, PAG generates progress awareness for each conversation turn and constructs a high-quality awareness dataset.

### 3.2 PHASE 1: PROGRESS AWARENESS GENERATION

Progress Awareness Generation (PAG) is designed to automatically synthesize a high-quality progress awareness dataset and equip the policy before RL. It contains (i) trajectory segmentation, (ii) awareness generation, (iii) model-aware verification, (iv) diversity-preserving augmentation, and (v) model warm-up.

**Trajectory Segmentation.** Given a multi-turn dataset $\mathcal{D}$ containing trajectories formatted as in Eq. (2), we segment each trajectory $\tau$ at each assistant response and only those where the assistant's response is a function call will be retained. Each segmentation becomes an instance:

$$\tau' = (info^{\text{meta}}, c^{\text{his}}, q, a^{\text{fc}}, c^{\text{fut}}),$$

where $info^{\text{meta}}$ contains tool descriptions/schema and the scenario description, $c^{\text{his}}/c^{\text{fut}}$ are the conversation contexts before/after the current step, $q$ is the current user query, and $a^{\text{fc}}$ is the ground-truth function call for this step.[1]

**Awareness Generation.** Given these segmented instance, an off-the-shelf generator $LLM_{\text{gen}}$ is employed to generate a compact textual *awareness document* $S^a$ for each $\tau'$:

$$S^a = LLM_{\text{gen}}(\tau'; \text{prompt}),$$

where the prompt elicits three components: (i) a concise *history summary* capturing user intent, function calling history, and important arguments; (ii) a short *future plan*, including anticipated function sequence and decision points; (iii) minimal *rationale* that links history to plan. We denote the raw corpus as $\mathcal{D}^{\text{raw}} = \{S^a\}$.

**Model-Aware Verification.** Although $LLM_{\text{gen}}$ is typically a highly capable LLM, the quality of the generated awareness still requires validation. At this stage, based on the principle that *an ideal progress awareness should contain the necessary information to reconstruct the answer*, we introduce a Model-Aware Verification operation. In this stage, a frozen copy of the target policy is employed as $LLM_{\text{ver}}$. It attempts to recover the function call solely from $S^a$ in the absence of the original conversation:

$$\hat{a}^{\text{fc}} = LLM_{\text{ver}}(S^a, info^{\text{meta}}),$$

Subsequently, a normalized equivalence predicate $\text{Eq}(\cdot, \cdot)$ checks schema-level equality (argument order invariance, whitespace-insensitive strings, commutative sets/lists):

$$\text{Eq}(\hat{a}^{\text{fc}}, a^{\text{fc}}) = \mathbb{I}[\text{schema\_equal}(\hat{a}^{\text{fc}}, a^{\text{fc}})],$$

only instances with $\text{Eq} = 1$ are retained, yielding $\mathcal{D}^{\text{ver}} = \{S^a \in \mathcal{D}^{\text{raw}} \mid \text{Eq}(\hat{a}^{\text{fc}}, a^{\text{fc}}) = 1\}$.

---

[1]Details of $\mathcal{D}$ construction are provided in the Appendix A.3.

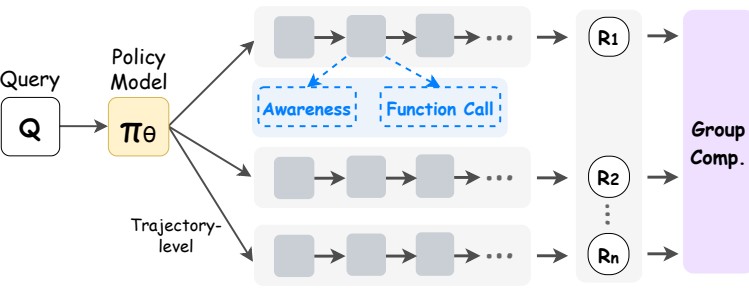

Figure 3: Overview of PAG-RL. By incorporating progress awareness, the policy model generates multi-step action trajectories in response to a query.

**Diversity-Preserving Augmentation.** To mitigate lexical overfitting and improve robustness, an augmenter $LLM_{\mathrm{aug}}$ is applied to perform semantic-level transformations over each verified $S^a$ with a randomly sampled operation type $\in$ {paraphrase, schema-perturb, word-mask}:

$$\tilde{S}^a = LLM_{\mathrm{aug}}(S^a; \text{type}),$$

yielding $\mathcal{D}^{\mathrm{aug}} = \{\tilde{S}^a\}$. Specifically, the instructions for each augmentation operation are as follows: (1) paraphrase: paraphrasing user intents to introduce lexical and syntactic variation; (2) schema-perturb: modifying function names and parameter values within permissible ranges to simulate realistic perturbations; (3) word-mask:applying random masking to function-related word to enhance robustness to incomplete or noisy inputs.

Consequently, a high-quality dataset containing strong progress awareness $\mathcal{D}^{\mathrm{aug}}$ is obtained.

**Model Warm-Up.** To facilitate high-quality progress awareness generation during the next phase, we employ the aforementioned $\mathcal{D}^{\mathrm{aug}}$ for model warming up. In order to enhance the familiarity with function call structure, a lightweight cold-start dataset $\mathcal{D}^{\mathrm{cs}}$ is additionally curated, extracted from the original training set $\mathcal{D}$ and adhering well-formed function-call exemplars (no awareness text). The final SFT dataset is given by $\mathcal{D}^{\mathrm{sft}} = \mathcal{D}^{\mathrm{aug}} \cup \mathcal{D}^{\mathrm{cs}}$.

For awareness instances, the input is $(info^{\mathrm{meta}}, c^{\mathrm{his}}, q)$ and the label is $\tilde{S}^a$, with the SFT loss as:

$$\mathcal{L}_{\mathrm{sft}}(\theta) = -\sum_{(\cdot, \tilde{S}^a) \in \mathcal{D}^{\mathrm{aug}}} \log \pi_\theta(\tilde{S}^a \mid info^{\mathrm{meta}}, c^{\mathrm{his}}, q) - \sum_{(\cdot, a^{\mathrm{fc}}) \in \mathcal{D}^{\mathrm{cs}}} \log \pi_\theta(a^{\mathrm{fc}} \mid info^{\mathrm{meta}}, c^{\mathrm{his}}, q).$$

As a result of this stage, $\pi_{\theta'}$ is obtained with a learned ability to summarize history and outline a plan for future, namely aforementioned *progress awareness*.

### 3.3 PHASE 2: PROGRESS AWARENESS-GUIDED RL

After strengthening the LLM's progress awareness via the PAG stage, we introduce Progress Awareness Guided RL (PAG-RL), which explicitly incorporates progress awareness into end-to-end reinforcement learning, with the goal of improving the model's effectiveness in realistic scenarios. This section will be organized with an *awareness-guided rollout* and a *composite reward*, and *optimization procedure*.

#### 3.3.1 AWARENESS-GUIDED ROLLOUT

Following the formulation in Sec. 3.1, the state $S_{i,j}$ includes all prior interactions, including the user query $Q_i$, the sequence of all past actions $\{A_{\leq j}\}$, and corresponding observations $\{O_{\leq j}\}$ in rollout. At each intra-turn step $(i, j)$, instead of conditioning on the entire raw prefix, $\pi_{\theta'}$ first emits a compact progress awareness as:

$$S_{i,j}^a \sim \pi_{\theta'}(\cdot \mid info^{\mathrm{meta}}, Q_i, \{(A_{\leq j-1}, O_{\leq j-1})\}), \tag{3}$$

---

**Algorithm 1** Progress-Awareness-Guided RL (one training iteration)

---

**Inputs:** warmed-up policy $\pi_{\theta'}$, env $E$, group size $G$

1: Initialize $\pi_\theta \leftarrow \pi_{\theta'}$
2: **for** $\ell \in \{1, \dots, L\}$ **do**
3:     Sample trajectory $\tau_\ell$ with $S_{i,j}^a \sim \pi_\theta(\cdot \mid \text{context})$, $A_{i,j} \sim \pi_\theta(\cdot \mid S_{i,j}^a)$
4: **end for**
5: $\hat{A}_\ell \leftarrow \frac{R(\tau_\ell) - \mu_R}{\sigma_R}$ for $\ell = 1, \dots, L$                          ▷ advantages
6: Update $\pi_\theta$ via GRPO objective with advantages $\{\hat{A}_\ell\}$       ▷ **progress-aware** optimization

---

Conditioned on $S_{i,j}^a$, $\pi_{\theta'}$ then generates the action $A_{i,j+1}$ in a chain-of-thought (CoT) style:

$$A_{i,j+1} \sim \pi_{\theta'}(\cdot \mid \textit{info}^{\text{meta}}, Q_i, S_{i,j}^a), \tag{4}$$

$$= \texttt{<think>}\ a_{i,j+1}^{\text{think}}\ \texttt{</think>}\ \texttt{<answer>}\ a_{i,j+1}^{\text{fc}}\ \texttt{</answer>}, \tag{5}$$

where textual action $A_{i,j+1}$ includes intermediate reasoning $a_{i,j+1}^{\text{think}}$ and the function call $a_{i,j+1}^{\text{fc}}$. After $a_{i,j+1}^{\text{fc}}$ executed in $E$, $(O_{i,j}, r_{i,j})$ will be returned and the rollout proceeds. The rollout will terminate either upon all queries completed or reaching the maximum number of interactions, obtaining a trajectory $\tau$, after which the reward $R(\tau)$ of the trajectory is computed as defined by the corresponding Equ. 2.

### 3.3.2 REWARD DESIGN

Each action in rollout will receive a composite reward including textual structural validity, function schema correctness, task success, and execution efficiency, which can be formulated as:

$$r_{i,j} = \alpha_{\text{fmt}} \underbrace{\mathbb{I}\big[\text{TEMPLATE}(A_{i,j})\big]}_{r^{\text{fmt}}} + \alpha_{\text{schema}} \underbrace{\mathbb{I}\big[\text{SCHEMA}(A_{i,j})\big]}_{r^{\text{schema}}} + \alpha_{\text{acc}} \underbrace{\mathbb{I}\big[\text{SUCCESS}(S_{i,j}^E \Rightarrow S_{i,j+1}^E)\big]}_{r^{\text{acc}}} - \lambda \underbrace{\mathbb{I}[A_{i,j} \neq \emptyset]}_{r^{\text{pen}}}, \tag{6}$$

with $\alpha_{\text{fmt}}, \alpha_{\text{schema}}, \alpha_{\text{acc}}, \lambda > 0$. TEMPLATE enforces the output tags in Equ. 4; SCHEMA validates function name, argument values and types; SUCCESS checks if overall task is completed. If the executed function call satisfies the current user query and drives the environment into the correct target internal state, a positive reward is granted as $r^{\text{acc}}$. $r^{\text{pen}}$ regulate the overall length of the rollout trajectory, encouraging the policy to complete tasks efficiently during training.

### 3.3.3 OPTIMIZATION PROCEDURE

After obtaining multi-turn trajectories through rollout and computing the corresponding trajectory-level rewards based on the composite reward function, the Group Relative Policy Optimization (GRPO (Shao et al., 2024b)), a critic-free variant of PPO, is applied to optimize the policy.

Given a batch of $L$ trajectories $\{\tau_\ell\}_{\ell=1}^L$ rolled out by $\pi_{\text{old}}$, we compute the scalar return $R(\tau_\ell)$ and normalize it within the batch:

$$\hat{A}_\ell = \frac{R(\tau_\ell) - \mu_R}{\sigma_R}, \quad \mu_R = \frac{1}{L}\sum_\ell R(\tau_\ell), \quad \sigma_R^2 = \frac{1}{L}\sum_\ell (R(\tau_\ell) - \mu_R)^2.$$

Following GRPO, the same normalized advantage $\hat{A}_\ell$ is evenly assigned to all tokens in $\tau_\ell$ in an concated textual action representation:

$$A_{i,j}^{\text{con}} = \texttt{<sum>}S_{i,j}^a\texttt{</sum><think>}a_{i,j}^{\text{think}}\texttt{</think><answer>}a_{i,j}^{\text{fc}}\texttt{</answer>}, \tag{7}$$

Let $\tau_{\ell,(t)}$ be the $t$-th token and $\tau_{\ell,<t}$ its prefix (which includes $\texttt{<sum>}$ $S^a$, $\texttt{<think>}$, and $\texttt{<answer>}$ regions). The objective is

$$J_{\text{GRPO}}(\theta) = \frac{1}{L}\sum_{\ell=1}^L \frac{1}{|\tau_\ell|}\sum_{t=1}^{|\tau_\ell|} \min\left( \frac{\pi_\theta(\tau_{\ell,(t)} \mid \tau_{\ell,<t})}{\pi_{\text{old}}(\tau_{\ell,(t)} \mid \tau_{\ell,<t})}\hat{A}_\ell,\ \text{clip}\Big[\tfrac{\pi_\theta}{\pi_{\text{old}}}, 1-\epsilon, 1+\epsilon\Big]\hat{A}_\ell \right) - \beta_{\text{KL}}\, \text{KL}\big[\pi_\theta \parallel \pi_{\theta'}\big], \tag{8}$$

where the optional KL term regularizes the policy towards the SFT reference $\pi_{\theta'}$ to prevent reward hacking; $\epsilon$ and $\beta_{\text{KL}}$ are hyperparameters.

Table 1: Performance of PROGRA and other methods across different benchmarks and LLMs. Within each group, the **bolded** values denote the best performance, while the underlined values indicate the second-best performance. *Overall* denotes the average score across different categories.

| Model | Category | Method | BFCL | | | $\tau$-**Bench** | | |
|-------|----------|--------|---------|------|---------|---------|---------|--------|
| | | | *Overall* | *Base* | *Miss P.* | *Overall* | *Airline* | *Retail* |
| Qwen2.5-7B | Baseline | Base Model | 8.25 | 9.50 | 7.00 | 16.60 | 8.00 | 25.20 |
| | Prompting | Reasoning | 9.25 | 12.00 | 6.50 | 18.00 | 10.00 | 26.00 |
| | Training | SFT | 9.25 | 11.50 | 7.00 | 21.30 | **20.00** | 22.61 |
| | RL-based | MT-GRPO | 9.17 | 11.17 | 7.17 | 19.00 | 11.33 | 26.67 |
| | RL-based | **PROGRA** | **10.33** | **13.00** | **7.67** | **23.91** | **20.00** | **27.83** |
| xLAM-2-3B | Baseline | Base Model | 60.50 | 66.50 | 54.50 | 25.45 | 24.00 | 26.90 |
| | Prompting | Reasoning | 61.25 | 67.50 | 55.00 | 22.85 | 26.00 | 21.70 |
| | Training | SFT | 61.00 | 67.00 | 55.00 | 23.30 | 24.00 | 22.61 |
| | RL-based | MT-GRPO | 60.83 | 66.67 | 55.00 | 26.35 | 24.00 | 28.70 |
| | RL-based | **PROGRA** | **61.42** | **67.67** | **55.17** | **31.52** | **30.00** | **33.04** |
| xLAM-2-8B | Baseline | Base Model | 69.25 | 74.75 | 63.75 | 46.70 | 35.20 | 58.20 |
| | Prompting | Reasoning | 67.67 | 72.83 | 61.83 | 50.10 | **42.00** | 58.20 |
| | Training | SFT | 69.50 | 75.00 | 64.00 | 46.57 | 34.00 | 59.13 |
| | RL-based | MT-GRPO | 69.42 | 74.67 | 64.17 | 49.52 | 39.33 | 59.71 |
| | RL-based | **PROGRA** | **70.08** | **75.67** | **64.50** | **51.85** | 38.00 | **65.70** |

## 4 EXPERIMENTS

### 4.1 RESEARCH QUESTIONS

To rigorously evaluate the effectiveness of PROGRA, we propose the following research questions:

**RQ1:** Does PROGRA consistently outperform alternative training strategies across diverse datasets and backbone LLMs?

**RQ2:** What is the relative contribution of each stage and component of PROGRA to the overall performance gains (ablation study)?

**RQ3:** To what extent can models trained with progress-awareness data demonstrate enhanced capability in progress-guided decision-making?

**RQ4:** How well does PROGRA perform in real multi-turn conversation scenarios (case study)?

### 4.2 EXPERIMENT SETUP

**Benchmark & Evaluation.** We evaluate the performance of PROGRA on two widely used multi-turn function calling benchmarks: BFCL-V3 Multi-Turn (BFCL) (Patil et al., 2024) and $\tau$-Bench (Yao et al., 2024). For BFCL, we adopt two subsets: *Base* and *Miss Parameters* (*Miss. P*), which respectively provide a standard and an augmented evaluation setting. We follow the benchmark's official metric, *Executable Function Accuracy*. For $\tau$-Bench, we evaluate on the *airline* and *retail* scenarios. GPT-4o is used as the user simulator. Each evaluation sample is tested across 3 trials and averaged results reported.

**Backbone LLMs & Baseline.** We adopt Qwen2.5-7B-Instruct (Team, 2024), xlam2-3B and xlam2-8B (Prabhakar et al., 2025a) as backbone LLMs. These models span different sizes and architectures while exhibiting strong performance in function calling. We compare PROGRA against representative inference and training strategies designed to improve multi-turn function calling: (1) **Reasoning:** Enhances function call accuracy by prompting the model to generate chain-of-thought (CoT) reasoning. (2) **SFT:** Trains the model via supervised fine-tuning on multi-turn conversation data. (3) **MT-GRPO:** Vanilla multi-turn GRPO algorithm, following the RAGEN (Wang et al., 2025b) framework, which computes trajectory-level token advantages and applies the same reward setting as PROGRA.

Table 2: Ablation study showing incremental improvements from each component. '$\Delta$ Avg.' indicates the improvement over the base model after averaging the *overall*. 'w/o' means excluding this phase from training. 'PAG+MT-GRPO' indicates replacing PAG-RL with vanilla multi-turn GRPO.

| Model | Method | BFCL | | | $\tau$-Bench | | | $\Delta$ Avg. |
|---|---|---|---|---|---|---|---|---|
| | | *Overall* | *Base* | *Miss P.* | *Overall* | *Airline* | *Retail* | |
| Qwen2.5-7B | Base Model | 8.25 | 9.50 | 7.00 | 16.60 | 8.00 | 25.20 | - |
| | MT-GRPO | 9.17 | 11.17 | 7.17 | 19.00 | 11.33 | 26.67 | 13.3% |
| | w/o PAG-RL | 9.00 | 11.50 | 6.50 | 15.61 | 6.00 | 25.22 | -1.0% |
| | w/o PAG | 9.25 | 11.00 | 7.50 | 19.00 | 18.00 | 20.00 | 13.6% |
| | PAG + MT-GRPO | 9.00 | 10.75 | 7.25 | 20.48 | 14.00 | 26.95 | 18.6% |
| | **PROGRA** | **10.33** | **13.00** | **7.67** | **23.91** | **20.00** | **27.83** | 37.7% |
| xLAM-2-3B | Base Model | 60.50 | 66.50 | 54.50 | 25.45 | 24.00 | 26.90 | - |
| | MT-GRPO | 60.83 | 66.67 | 55.00 | 26.35 | 24.00 | 28.70 | 1.4% |
| | w/o PAG-RL | 60.50 | 66.50 | 54.50 | 25.91 | 24.00 | 27.83 | 0.5% |
| | w/o PAG | 61.00 | 66.83 | **55.17** | 26.04 | 26.00 | 26.09 | 1.3% |
| | PAG + MT-GRPO | 60.92 | 66.83 | 55.00 | 28.53 | 28.66 | 28.40 | 4.1% |
| | **PROGRA** | **61.42** | **67.67** | **55.17** | **31.52** | **30.00** | **33.04** | 8.1% |
| xLAM-2-8B | Base Model | 69.25 | 74.75 | 63.75 | 46.70 | 35.20 | 58.20 | - |
| | MT-GRPO | 69.42 | 74.67 | 64.17 | 49.52 | 39.33 | 59.71 | 2.6 % |
| | w/o PAG-RL | 67.33 | 73.00 | 61.67 | 50.70 | **44.00** | 57.39 | 1.8% |
| | w/o PAG | 69.00 | 74.50 | 63.50 | 50.44 | 40.00 | 60.87 | 3.0% |
| | PAG + MT-GRPO | 67.50 | 73.00 | 62.00 | 46.84 | 37.77 | 55.90 | -1.4% |
| | **PROGRA** | **70.08** | **75.67** | **64.50** | **51.85** | 38.00 | **65.70** | 5.1% |

**Training Details.** Training data is constructed following benchmark-specific pipelines. For BFCL, we generate 200 samples using the APIGEN-MT synthesis pipeline. Detailed implementation and prompts are provided in Appendix A.3. For $\tau$-Bench, we randomly sample 200 instances from the publicly available APIGEN-MT-5K dataset (Prabhakar et al., 2025c).

For reinforcement learning, we implement PROGRA using RAGEN, a public framework for multi-turn LLM agent training. Benchmark-provided finite-state machines serve as $E$ in Sec. 3.1. GRPO is adopted as the optimization method. We set the batch size to 8, GRPO group size to 8 and each sample is allowed up to 10 actions. Additional training details are reported in Appendix A.5.

### 4.3 EFFECTIVENESS OF PROGRA ACROSS DATASETS AND BACKBONE LLMS (RQ1)

We evaluate PROGRA against existing training strategies on three backbone LLMs across two benchmarks as Table 1, revealing the following key observations: 1) PROGRA consistently outperforms all baselines (e.g., SFT, MT-GRPO) across benchmarks and LLMs. 2) On BFCL, PROGRA improves performance by up to 25.21% on Qwen2.5-7B-Instruct; on $\tau$-Bench, it achieves a 44.05% gain on the same backbone. 3) On xLAM-2 models, PROGRA yields further improvements, with gains of 23.85% on xLAM-2-3B and 11.02% on xLAM-2-8B on $\tau$-Bench. Overall, these results demonstrate that PROGRA consistently enhances performance across diverse datasets and model architectures, underscoring its robustness and generalization capability in multi-turn function calling.

### 4.4 CONTRIBUTION OF INDIVIDUAL COMPONENTS (RQ2)

To examine the contribution of each stage and component of PROGRA, we conduct detailed ablation studies on BFCL and $\tau$-Bench with three backbone LLMs. The results are summarized in Table 2.

**Overall Effectiveness.** Across all models and benchmarks, complete PROGRA consistently outperforms, highlighting the essential role of integrating progress-awareness training with RL.

**Impact of Removing Components.** Table 2 shows that excluding either PAG or PAG-RL consistently lowers both BFCL and $\tau$-Bench scores across model scales. The drop is more pronounced when removing PAG-RL, underscoring that RL contributes the largest share of the improvement, while PAG alone also provides steady gains. Replacing PAG-RL with vanilla multi-turn GRPO yields only partial benefits, confirming the necessity of our tailored reinforcement stage.

**Comparison with MT-GRPO.** Using only PAG-RL does not surpass the performance of the vanilla MT-GRPO. However, when comparing PAG + MT-GRPO against the full PROGRA, we ob-

serve that the progress awareness introduced by PAG substantially enhances the LLM's capabilities. Moreover, continuing with PAG-RL leads to even greater improvements. For example, on $\tau$-Bench with xLAM-2-8B, replacing PAG-RL with MT-GRPO results in a 10.69% performance drop.

### 4.5 CAPABILITY OF PROGRESS-AWARENESS GUIDANCE (RQ3)

To validate the impact on the improvement of progress awareness after each training phase in PROGRA, we conducted experiments on the reserved $\tau$-Bench validation set using Qwen2.5-7B-Instruct after different phases. In this experiment, Qwen2.5-7B-Instruct was tested in three forms: untrained, Phase-1 trained, and Phase-1+2 trained, generating conversation awareness at each stage (Base Aw., Phase-1 Aw., Phase-2 Aw.). Each variant also served as a verifier to assess awareness quality. "Aw." refers to the awareness generator and "Act." to the action verifier, while "Base/Phase-1/Phase-2" indicates the model's training stage (untrained, post-Phase-1, and post-Phase-2). The results show that, with the same Base Act. model, progressive training improves progress awareness quality, highlighting how each stage in PROGRA enhances the model's summarization and planning abilities.

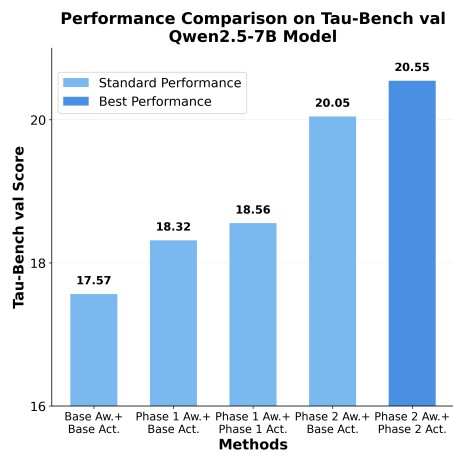

Figure 4: Awareness capability across phases

### 4.6 PRACTICAL CASE STUDY: MULTI-TURN CONVERSATION PERFORMANCE (RQ4)

To assess the practical effectiveness of PROGRA, we began with an airline booking scenario in $\tau$-Bench. **User Request:** *"Hi! I'd like to make some changes to my upcoming trip from New York to Chicago... I want to upgrade to economy class, add 3 checked bags, and change the passenger name to myself, Omar Rossi."* Steps marked with PA are explicitly guided by *progress awareness*.

| Direct Inference (Baseline) | PROGRA (Ours) |
|---|---|
| **Process:** | **Process:** |
| • Assumes details without verification | 1. Protocoling: request user ID & reservation ID  PA |
| • Skips user/booking checks | 2. Factual retrieval: |
| • Pure direct reasoning | `get_user_details`, `get_reservation_details` |
| | 3. Progress-aware reasoning (summary + plan)  PA |
| | 4. Early stop: inform no change needed  PA |

In this trajectory, the PROGRA-trained model proactively summarized the dialogue, capturing both the user request and earlier context (e.g., flight numbers, passenger details) to support accurate modifications. In contrast, the untrained model failed to summarize and relied only on the current step's flight number, leading to repetitive function calls.

Building on this case study, we further evaluate the two methods on 50 $\tau$-Bench dialogues. As summarized in Table 3, PROGRA consistently achieves higher accuracy with fewer steps compared to Direct inference, confirming its effectiveness in multi-turn conversation scenarios.

Table 3: Quantitative comparison.

| Method | Acc. | Avg. Steps | Risk |
|---|---|---|---|
| Direct | 0.26 | 27.08 | High |
| PROGRA | 0.40 | 23.30 | Low |

## 5 CONCLUSION

In this paper, we propose PROGRA, a novel training framework that introduces task progress awareness into the multi-turn function calling. We first identify the performance bottleneck of multi-turn function calling as the LLM's lack of overall task progress awareness. Based on this, we design a two-phase training process. In Phase 1, an automated data synthesis process is used to enhance the LLM's task awareness. In Phase 2, progress awareness is integrated into end-to-end multi-turn reinforcement learning, significantly improving the LLM's performance in multi-turn function calling. The performance improvements achieved exceed those of existing training strategies across two public datasets and three backbone LLMs.

ETHICS STATEMENT

We have thoroughly assessed the potential societal and ethical implications of our work, including the risk of misuse, unfair bias, and broader negative impacts. We affirm that this research adheres to the ethical standards set forth by the ICLR Code of Ethics.

REPRODUCIBILITY STATEMENT

To ensure transparency and reproducibility of our research, we provide all necessary details to enable independent replication of our results. We provide our core codes in an anonymized repository `https://anonymous.4open.science/r/Progra_ICLR26-57F0`. All hyperparameters, configurations, and training procedures are reported in the main text or in Appendix A.5.

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

# A APPENDIX

## A.1 LIMITATIONS AND FUTURE WORK

In this work, we propose PROGRA, a training paradigm that incorporates progress awareness into enhancing multi-turn function calling capability of LLM, achieving superior performance over other training strategies on two public benchmarks. Despite its promising results, the idea of PROGRA has not yet been validated with other reinforcement learning algorithms, and during rollout training, PROGRA incurs additional time costs by requiring one more inference per action compared to vanilla multi-turn reinforcement learning. In future work, we plan to extend the application of PROGRA to different reinforcement learning algorithms, and to address its extra computational overhead, we will explore lightweight awareness generation modules to reduce the cost.

## A.2 THE USE OF LARGE LANGUAGE MODELS (LLMs)

We used an LLM exclusively for grammar, wording, and stylistic refinement of the manuscript. The LLM did not generate scientific claims, perform analyses, design experiments, select or interpret results, or introduce new references. Following any language suggestions, we conducted extensive manual review and cross-checking to prevent hallucinations or factual inaccuracies. All substantive content and conclusions are authored and verified by the authors; any remaining errors are our own.

## A.3 DATA SYNTHESIS

We provides a detailed account of the data synthesis methodology employed in this study. The process is adapted from the apigen-mt framework (Prabhakar et al., 2025b), which comprises a two-phase approach: (1) task configuration and groundtruth generation, and (2) human-agent-environment interaction trajectory collection. In the initial phase, an objective, its corresponding function call, and the resulting output are generated for each data instance. This generation is guided by provided APIs, a set of predefined rules, and a specified domain. Subsequently, each generated instance undergoes a rigorous verification protocol that assesses its syntactic formatting and operational executability. A majority voting mechanism, arbitrated by an LLM, is then employed to ascertain the correctness of the data. Should an instance fail these verification checks or the majority vote, the model will analyze the failure cases, formulate a corrective strategy, and reiterate the generation-verification cycle until the data successfully meets all validation criteria. Following successful validation in the first phase, the synthesized data is deployed into a simulated human-agent-environment to produce an interaction trajectory via rollout. This resultant trajectory is then compared against the groundtruth trajectory established in the initial phase. Only those instances where the two trajectories exhibit exact correspondence are retained for the final synthetic dataset.

The data synthesis pipeline in this study is adapted from the apigen-mt framework, with significant modifications to four core stages: initial configuration generation, query generation, action generation, and correctness verification. The following sections detail these customized processes.

**Initial Config Generation**: The process commences with the generation of an initial_config, which serves as the foundational state for each dialogue trajectory. To ensure contextual relevance, this stage is seeded with data from a predefined JSON dataset. Specifically, a subset of data entries is first filtered based on a specified involved_class criterion. From this filtered subset, a random selection of existing initial_config instances is sampled. These sampled configurations act as reference templates or exemplars. A generative model then synthesizes a new, distinct initial_config that adheres to the structural and schematic patterns of the references. This targeted sampling strategy ensures that the generated initial states are not only well-formed but also thematically aligned with the desired domain, thereby enhancing the relevance of the subsequent dialogue synthesis.

**Query Generation**: The dialogue synthesis process begins each interaction cycle with a query generation step. In this phase, the system leverages the complete preceding context—comprising the initial_config and the historical dialogue trajectory—as input. Conditioned on this context, a large language model (LLM) is invoked via a single API call to produce a new query. The design of this step ensures that each generated query is a logical and progressive continuation of the interaction. Queries are formulated to be explicit requests for environmental modification that depend on the out-

comes of prior turns, thus establishing a coherent and causally linked chain of reasoning throughout the dialogue.

**Action Generation**: Following each query generation turn, the pipeline proceeds to synthesize a corresponding action. In this phase, the query formulated in the immediately preceding turn is supplied to an LLM prompted specifically for action synthesis. To mitigate generation errors and enhance the reliability of the output, a distributed majority voting mechanism is employed. This is implemented by triggering multiple, parallelized API calls to the LLM to produce a set of candidate actions. These candidates are then subjected to a consensus protocol, where votes for each unique candidate are aggregated. The action that surpasses a predefined frequency threshold is selected as the definitive result. A critical constraint is that all generated actions must conform to a strict, machine-parsable format, such as a list of function calls (e.g., [func_name1(arg1=value1,...), func_name2(...)]). If no candidate action achieves the required consensus threshold, the generation process for the current trajectory is aborted to prevent the inclusion of low-confidence or ambiguous data.

**Correctness Verification**: The data verification process is a rigorous, automated pipeline designed to ensure the functional and semantic correctness of multi-turn AI agent trajectories. For each trajectory, a hermetic execution environment is instantiated from an initial configuration to guarantee reproducible validation. The pipeline then employs a two-tiered verification protocol for each turn: first, it deterministically checks if an agent's action causes an observable state transition in the environment. If no change is detected (a common result for read-only operations), a secondary check uses a majority-voting consensus from a Large Language Model (LLM) to adjudicate the action's semantic validity based on the user's query. Crucially, the process adheres to a strict sequential policy, terminating immediately upon the first failed turn to ensure that only fully coherent and causally valid interaction sequences are retained in the final dataset.

This data synthesis methodology offers several distinct advantages that collectively enhance the quality, relevance, and reliability of the generated dataset. By seeding the process with domain-specific exemplars, the pipeline ensures that all synthesized trajectories are thematically relevant and contextually grounded from their inception. The iterative, context-aware generation of queries and actions promotes the creation of coherent, multi-turn dialogues that exhibit logical and causal consistency. Furthermore, the integration of a consensus-based validation mechanism for action generation significantly improves the accuracy and reliability of the output by filtering out erroneous or low-confidence predictions. Crucially, the final execution-based verification stage provides a rigorous guarantee of functional fidelity, ensuring that every data point corresponds to a verifiable and correct interaction within the target environment. This multi-layered approach to generation and validation yields a high-quality dataset that is not only syntactically sound and semantically coherent but also empirically validated for functional correctness.

## A.4 PROMPTS

The prompts we use when synthesizing data include the prompt for the -Initial Config Generation(Fig. 5), Task Generation(Fig. 6), Action Generation(Fig. 7).

| **Initial Config Generation Prompt** |
|---|
| You are an environment generator. Your task is as follows: you will be provided with several examples of environment initial configurations, along with some notes. Based on these, generate **ONE** new environment initial configuration. The content is entirely up to you, but the overall format should be inspired by the examples (not an exact copy). Your output must be a valid, directly parsable JSON string. You **MUST NOT** include any extra text, explanations, or JSON hints outside of the JSON itself. You only need to output ONE new configuration. |

Figure 5: Initial Config Generation Prompt.

---

**Task Generation Prompt**

You are a helpful assistant, which will generate a trajectory containing Queries and Actions. You will be provided with a basic description of an existing scenario and an introduction to the tools available in that scenario.

Your task is to output a reasonable and clear query based on the result of the previous message. If the previous message represents an action or the initialization of a trajectory (no previous message), you must output a Query that can be solved by calling the tools within the environment. Your output query must involve a change to the environment state (e.g., adding or moving files, modifying content and so on).

There are some important notes you should follow.

1. The query should be progressive, where each query depends on the successful answer of the previous one, forming a realistic problem-solving process.

2. The query should not be too complex, it is better to cost 2-4 function calls to complete the query.

3. Please ensure that the Query you output is very clear and explicit, and that it allows only one possible solution.

4. Your query should involve a change to the environment state, try not only to display information.

Here is the initial config: {initial_config}

---

Figure 6: Task Generation Prompt.

---

**Action Generation Prompt**

You are a helpful assistant, which will generate a trajectory containing Queries and Actions. You will be provided with a basic description of an existing scenario and an introduction to the tools available in that scenario.

Your task is to output a correct function call output based on the result of the previous message and query. Your output should include some function calls that strictly follow the required format. The functions you call must come from the provided tool descriptions. The function call should be the following format:

[func_name1(arg_name1=value1,arg_name2=value2...),
func_name2(arg_name1=value1,arg_name2=value2...)...]

---

Figure 7: Action Generation Prompt.

Here are the prompts we used in Progra for generating progress awareness in PAG (Fig. 8), and Optimization in RAG-RL (Fig. 9)

---

**Progress Awareness Generation Prompt**

You are a help assistant responsible for summarizing important information based on the history of the conversation and providing a plan for invoking the correct function calls. Carefully analyze the current multi-turn conversation and generate a detailed summary (and helpful plan). **Do not directly output any function calls**, just output your summary in text format.

In your summary, ensure the following points are clearly addressed:

1. **User's Needs and Intent**: Please analyze the provided historical information and accurately identify the user's needs and goals. Summarize the content in a brief and clear manner to ensure that any subsequent work can fully understand the user's requirements and objectives based on your summary.

2. **Extracted Parameters**: Please list all relevant parameters mentioned or clarified in the conversation, ensuring complete understanding. Only include parameters that have been explicitly confirmed, and avoid making assumptions or guessing values for information that has not been clearly stated. If the historical information provided lacks any parameters necessary to meet the user's needs, confirm and point them out.

---

3. **Function Call History**: Carefully review previous function calls made, including whether they were successful or failed, and identify any potential issues. Ensure you clearly summarize the results of previous function calls without making assumptions about the context or next steps.

4. **Environment State Awareness**: Review and summarize the entire conversation so far, paying special attention to previous function calls and their results. In each turn of history, the environment sequentially executes the functions output in last turns. If a function fails, the subsequent ones won't be executed, but the successful ones will affect the environment state. In the current turn, you must reason the current environment state based on the executed functions and their results in last turns. You need to fully understand the current environmental state in order to make the correct choices. If you cannot understand it, you cannot make unreasonable assumptions about the state.

5. **Future Planning**: Based on the current and prior conversation, propose a clear action plan. Avoid speculation; instead, provide a plan that logically follows from the facts at hand. Do not directly output any function calls; instead output your plan in text format.

6. **Relevant Important Context**: Include any other context from the conversation that could aid in ensuring the next function call is accurate and appropriate. This can include non-explicit user preferences, hints from previous statements, or details that, while not directly related to the goal, may still influence the next action.

Ensure that your summary is detailed and comprehensive, clarifying any ambiguities and accurately reflecting the history of the conversation to allow the next response or action to be as accurate and informed as possible.

Here is the history of prior conversations and current user query.

**History of Conversation**

{current_history_str}

**Current User Query**

{query}

Please make a summary based on the above conversation and system prompt.

Figure 8: Progress Awareness Generation Prompt.

**Optimization Prompt**

You are an expert in invoking functions. You will be given a summary of a dialogue between a user and an AI assistant, as well as a set of available functions. Based on the summary and the questions posed by the user, you must select and call functions to achieve the user's goal. You must return the response in the following format:

⟨ summary ⟩
Summarise the dialogue so far for subsequent reasoning
⟨ /summary ⟩
⟨ think ⟩
Express your thought process. In this section, you should carefully consider the question's intent, your reasoning for selecting the appropriate function to call, and how you plan to fill in the function's parameters and arguments. Reflect on the user's needs to ensure the correct choice is made.
⟨ /think ⟩
⟨ answer ⟩
Output function call in the following format:
[func_name1(arg_name1=value1,arg_name2=value2...),
func_name2(arg_name1=value1,arg_name2=value2...)...]
⟨ /answer ⟩
- You MUST NOT include any text other than the ⟨summary⟩, ⟨ think⟩ and ⟨answer⟩ sections. Follow the provided format strictly. - **You MUST call only the functions provided in the document**. The function names and parameters 'func_name1', 'func_name2' and 'params_name1' shown in the ⟨answer⟩⟨/answer⟩ section are placeholders used only to demonstrate the required output format. They are not actual function names to be called. Please select and use only from the real functions provided as possible. - When filling in the

parameters in the function call, replace params_value1 and params_value2 with the correct values as per the document. Ensure the parameters are properly named and formatted. - If additional parameters are needed to call the function correctly, output "I need more information." in the ⟨answer⟩⟨/answer⟩ section with a text format rather than a list-style answer. - Please strictly adhere to the list of functions provided to complete the task. These functionns may be similar to common commands you are familiar with (such as cd, touch, etc.), but you are strictly prohibited from using any functions that are not on the provided list to complete the task. - If you find that none of the functions in the provided list can directly accomplish the task, you must not attempt to use other available functions in a roundabout way to force the completion of the task. In that case, simply output the following in the ⟨answer⟩⟨/answer⟩ tags: there are no appropriate functions. - In each round, the environment sequentially executes the functions output by you in last turns. If a function fails, the subsequent ones won't be executed, but the successful ones will affect the environment state. In the next round, you must infer the current environment state based on the executed functions and their results in last turns, ensuring that failed calls are not executed again.

Figure 9: Optimization Prompt in PAG-RL

## A.5    Training Details

During the PAG data synthesis phase, we use GPT-4o as both $LLM_{gen}$ and $LLM_{aug}$ . For each data entry, we allow a maximum of 4 iterations, and employ 5 reviewers in APIGen-MT (Prabhakar et al., 2025a) to assess the executability and correctness of the generated data.

During the warm-up stage in PAG, we employed supervised fine-tuning (SFT) using the LoRA (Low-Rank Adaptation) approach. We conducted hyperparameter searches over the following ranges: the LoRA rank was varied from 8 to 16; the learning rate ranged from 1e-7 to 1e-6; the batch size was set between 8 and 16; and the LoRA $\alpha$ (alpha) parameter was searched within the range of 8 to 64. A held-out validation set was used to evaluate whether the large language model (LLM) had been successfully trained during this warm-up phase.

In the reinforcement learning stage, we adopted the GRPO algorithm to optimize the policy. A KL-divergence penalty with a coefficient of $\beta = 0.001$ was used. The training was performed with a batch size of 8 and 8 rollout trajectories per batch. The maximum sequence length was limited to 16,384 tokens, and the temperature for the rollout generation was set to 1. Each training experiment was limited at 200 iterations, with a maximum of 10 actions per rollout. We continued to use the LoRA-based training strategy during this phase. Hyperparameter tuning was conducted over the following ranges: LoRA rank between 8 and 16, LoRA $\alpha$ between 32 and 64, and learning rate from 1e-9 to 1e-5.

