# OpenReview forum: "Progra: Progress-Aware Reinforcement Learning for Multi-Turn Function Calling"
_ICLR.cc/2026/Conference — ICLR 2026 Conference Withdrawn Submission_

### Official Review · Reviewer_8yxm · 2025-10-31

**Soundness:** 2
**Presentation:** 3
**Contribution:** 2
**Rating:** 4
**Confidence:** 4

**Summary:**

This paper introduces PROGRA, a two-phase training framework designed to improve LLMs' multi-turn function calling capabilities by explicitly incorporating "progress awareness"—the ability to summarize conversation history and plan future actions. Phase 1 (PAG) automatically constructs a dataset where each conversation turn is augmented with generated summaries and plans, verified through model-aware validation. Phase 2 (PAG-RL) integrates this progress awareness into GRPO-based reinforcement learning, where the model first generates awareness text before producing function calls. Experiments on BFCL and τ-Bench with three backbone models (Qwen2.5-7B, xLAM-2-3B/8B) show improvements over baselines, with gains ranging from modest (1.5% on xLAM-2-3B BFCL) to substantial (44% on Qwen2.5-7B τ-Bench).

**Strengths:**

- Well-motivated problem: The observation that existing multi-turn training methods degrade to single-turn samples and neglect task-level planning is valid and clearly articulated. Figure 1 effectively illustrates the delayed feedback issue in multi-turn conversations.
- Comprehensive pipeline design: The PAG data synthesis pipeline with model-aware verification and diversity-preserving augmentation demonstrates solid engineering. The two-phase approach (PAG for data generation + PAG-RL for training) is systematic.
- Systematic evaluation: The paper tests on multiple models (3 backbone LLMs) and benchmarks (BFCL, τ-Bench), includes ablation studies (Table 2), and provides awareness capability analysis (Figure 4). The experimental design covers different model scales.

**Weaknesses:**

1. Questionable conceptual novelty: The core concept of 'progress awareness' appears to be a repackaging of existing ideas (history summarization + future planning). The paper mentions SPA-RL's 'stepwise progress attribution' in related work but fails to clearly differentiate PROGRA's contributions. Section 2.2 lists several recent multi-turn RL works without adequate comparison.
2. Model-Aware Verification creates circular dependency: Using the target policy πθ to verify generated awareness quality (Section 3.2) is fundamentally problematic. If the model is weak initially, verification will approve low-quality data, perpetuating poor training. The paper provides no analysis of verification failure rates, quality distribution, or strategies to break this cycle.
3. Inconsistent and limited experimental gains: Table 1 shows highly variable improvements ranging from 1.5% (xLAM-2-3B BFCL Overall) to 44% (Qwen2.5-7B τ-Bench), suggesting the method lacks robustness. More critically, Table 2 Row 'PAG + MT-GRPO' for xLAM-2-8B shows **negative** results (-1.4% compared to baseline), directly contradicting the hypothesis that PAG improves awareness. Only 200 training samples are extremely small. It is unclear if results generalize beyond these specific benchmarks.
4. Missing computational cost analysis: PAG-RL requires generating awareness text before each action, at least doubling inference cost. Section A.1 acknowledges 'additional time costs' but provides zero quantitative analysis (no training time, wall-clock hours, FLOPs, or token throughput comparisons). This is critical for practical deployment and represents a major gap.
5. Weak baselines and missing comparisons: The paper only compares against vanilla MT-GRPO as the RL baseline, despite mentioning ARTIST, RLFactory, SPA-RL, and other recent methods in Section 2.2. The 'Reasoning' baseline is merely CoT prompting. No comparison with simpler alternatives like longer context windows or retrieval-augmented generation.
6. Limited evaluation scope: Only evaluation on two benchmarks (BFCL, τ-Bench). Case study (Section 4.6) shows only 1 cherry-picked example from 50 samples, detailed error analysis is missing.

**Questions:**

1.	How do you address the circular dependency in Model-Aware Verification? If the target model $\pi_{\theta}$ is initially weak, won't it approve low-quality awareness data, leading to poor SFT and subsequent RL training? Can you provide verification accuracy curves or inter-annotator agreement with human labels?
2.	Table 2 shows 'PAG + MT-GRPO' performs worse than baseline on xLAM-2-8B (-1.4%). This directly contradicts your central claim that PAG improves awareness. Is this a statistical artifact, or does PAG harm certain model architectures? Please explain.
3.	Please provide quantitative analysis: (a) Total training time in GPU-hours (PROGRA vs MT-GRPO), (b) Inference latency per action with/without awareness generation, (c) Total tokens generated and associated API costs. Without this data, the practicality of your method cannot be assessed.
4.	Why only compare against vanilla MT-GRPO for RL? Can you add comparisons with ARTIST, RLFactory, or SPA-RL? Or at minimum, demonstrate that MT-GRPO is a strong baseline by comparing it against these methods?
5.	With only 200 training samples, how do you ensure results aren't overfitting to the small benchmarks?

---

> ### Author Response · Authors · 2025-11-25
>
> >**W1. Clarifying the Novelty and Contributions of Progra**
>
> Thank you very much for raising this concern, which allows us to better clarify our contributions.
>
> ##### Novelty and Contribution of Progra
>
> Significant progress has been made recently, but no multi-turn reinforcement learning (RL) algorithm has yet explicitly modeled, utilized, and optimized task progress awareness within the RL framework. Progra successfully fills this gap with the following two core components, which serve as its key contribution and differentiate it from other methods:
>
> 1. Progress Awareness Generation (PAG): This phase automatically synthesizes a **high-quality task progress-aware dataset** by summarizing conversation history and planning future task actions. Based on this generated dataset, the model is able to treat task progress as an ongoing, evolving process, ensuring alignment between historical context and future task objectives.
>
> 2. Progress Awareness Guided Reinforcement Learning (PAG-RL): To the best of our knowledge, this is the first RL-based method that **directly integrates progress awareness into the training algorithm**. Existing methods may simplify multi-turn RL into isolated decision steps (e.g., stepwise attribution or trajectory scoring), whereas PAG-RL ensures that each decision is made with a global view of task progress, which is crucial for achieving long-term goal alignment and avoiding redundant or incomplete actions.
>
> By systematically integrating this task-level awareness, Progra significantly outperforms other methods that optimize for single-turn task accuracy or rely on isolated decision-making, as evidenced by the experimental results in the paper (see Tables 1 and 2).

---

> > ### Author Response · Authors · 2025-11-25
> >
> > >**W1. Clarifying the Novelty and Contributions of Progra**
> >
> > ##### Comparison with Other Multi-turn RL Approaches
> > With due respect, we further explore the distinction between Progra and other multi-turn RL algorithms to identify the unique contributions of Progra.
> >
> > First, we conduct a comparison with SPA-RL.The core contribution of SPA-RL is its use of a **progress estimator** to compute the stepwise contribution of each action in a trajectory, i.e., how each action contributes to task completion, and to **assign rewards** accordingly for each step to help with training.
> >
> > With respect to this, we need to clarify the difference between SPA-RL and Progra, specifically regarding the **integration of progress awareness within the RL training process**. Although SPA-RL employs a stepwise progress attribution mechanism for reward redistribution, it does not explicitly model or guide large language models (LLMs) to understand the relationship between historical actions and future steps in multi-turn tasks. SPA-RL does not directly adjust the agent's behavior based on task progress during training and lacks the direct optimization of the progress awareness capabilities of the progress estimator.
> >
> > In contrast, Progra not only trains the strategy to **understand, summarize, and plan long-term trajectories** during the PAG phase, but also fully leverages this capability within the end-to-end RL training framework. Progra actively **converts progress awareness into guidance for the policy rollout** and directly optimizes the strategy's progress-awareness ability through the RL algorithm, reducing redundant actions and optimizing the alignment of local actions with overall task completion. This presents a fundamental difference from SPA-RL’s reward redistribution approach, which does not directly adjust the agent’s behavior based on progress awareness during training.
> >
> > Furthermore, the other multi-turn RL algorithms mentioned in Section 2.2 of our paper also lack the modeling, utilization, and optimization of task progress awareness as described above. For example, ARTIST\[1] and RLFactory\[2] emphasize end-to-end multi-turn RL algorithms but **do not model intermediate results or progress**, MT-PPO\[3] is limited to two-turn dialogue scenarios and its credit assignment rewards do not derive from the specific task's progress, and DMPO\[4] has not expanded its application to RL-based virtual reality (RLVR) tasks.
> >
> > In summary, Progra uniquely integrates task progress awareness into the multi-turn reinforcement learning framework through two core components: Progress Awareness Generation (PAG), which synthesizes task progress-aware data, and Progress Awareness Guided Reinforcement Learning (PAG-RL), which directly incorporates this awareness into the RL training process to optimize long-term goal alignment and reduce redundant actions. This sets it apart from other methods like SPA-RL, which only employs progress attribution without directly guiding the agent’s behavior based on task progress.
> >
> >
> > \[1] Singh, J., Magazine, R., Pandya, Y., & Nambi, A. (2025). Agentic reasoning and tool integration for llms via reinforcement learning. arXiv preprint arXiv:2505.01441.
> >
> > \[2] Chai, J., Yin, G., Xu, Z., Yue, C., Jia, Y., Xia, S., ... & Lin, W. (2025). Rlfactory: A plug-and-play reinforcement learning post-training framework for llm multi-turn tool-use. arXiv preprint arXiv:2509.06980.
> >
> > \[3] Zeng, S., Wei, Q., Brown, W., Frunza, O., Nevmyvaka, Y., Zhao, Y. K., & Hong, M. (2025, May). Reinforcing multi-turn reasoning in llm agents via turn-level credit assignment. In ICML 2025 Workshop on Computer Use Agents.
> >
> > \[4] Shi, W., Yuan, M., Wu, J., Wang, Q., & Feng, F. (2024). Direct multi-turn preference optimization for language agents. arXiv preprint arXiv:2406.14868.

---

> > > ### Author Response · Authors · 2025-11-25
> > >
> > > >**W2&Q1. Clarification on Model-Aware Verification and Quality Control**
> > >
> > > Thank you for raising this critical question. We clarify that Model-Aware Verification does not suffer from a "weak model endorsing itself" circular dependency, because the verification signal is determined by **external ground-truth answer matching** rather than by the model’s subjective approval.
> > >
> > > Concretely, during the verification stage, the initial policy $\pi_{\theta}$ first generates a progress-awareness content for the current conversation, including a summary of the conversation history and a plan for future function call paths. Then, $\pi_{\theta}$ is required to predict the next function call solely based on this progress-awareness content, **without accessing the original conversation**. We compare the predicted function call with the dataset-provided ground-truth answer: only if they match is the corresponding progress-awareness instance retained as a high-quality sample; otherwise it is discarded. Therefore, passing verification is not controlled by the model’s approval but by agreement with the true labels.
> > >
> > > This procedure imposes a strong constraint:
> > > 1. If $\pi_{\theta}$ can correctly produce the next function call using only a given progress-awareness content, that content must contain **all essential information needed** for the function call, e.g., the user goal, relevant historical state, critical arguments, and a coherent forward plan. We thus regard it as high-quality awareness.
> > > 2. Conversely, low-quality awareness typically **misses user intent or key parameters**. Since the model cannot revisit the raw dialogue, such omissions directly prevent the next-step call from matching the ground truth, and the instance is automatically filtered out. As a result, a weak policy cannot easily "approve" low-quality awareness, which mechanistically avoids circular dependency.
> > >
> > > Due to limited resources, we have not yet completed the experiment on the consistency between human annotation quality of the awareness data and the actual verification performance. We will complete this experiment as soon as possible.

---

> > > > ### Author Response · Authors · 2025-11-25
> > > >
> > > > >**W3,W5,Q2,Q4&Q5. Insights of Ablation Experiments, Comparison with More Baselines and Cross-Domain Experiments**
> > > >
> > > > Thank you for your concern. The models we selected, Qwen2.5-7B, xLAM-2-3B, and xLAM-2-8B, are commonly used in the function calling domain. Qwen2.5 is a general-purpose model, while the xLAM-2 series has been **specifically fine-tuned** for function calling tasks using publicly available APIGEN-MT dataset \[1], which naturally explains the strong basic performance of xLAM-2 models on Tau-Bench and BFCL.
> > > >
> > > > Despite the **differences in architecture** and task adaptability between Qwen2.5-7B and the xLAM models, the Progra method consistently demonstrates significant performance improvements across different model sizes and use cases, further validating its robustness. We will update additional experiments with other models during the rebuttal period to further verify the applicability and robustness of our approach.
> > > >
> > > > In the ablation study results presented in Table 2 of the paper, the gain from using only PAG training is generally smaller than the gain from other components. Furthermore, 'PAG + MT-GRPO' performs worse than the baseline on xLAM-2-8B (-1.4%). We believe this is due to a **misalignment between the training and evaluation objectives**. In the PAG training phase, we focus on enhancing the LLM's ability to summarize the current dialogue progress and plan future actions, without directly improving the LLM's multi-turn function calling capability. This subtle inconsistency in the training objectives leads to lower performance improvements and may occasionally even degrade actual performance (as noted in the data point you mentioned).
> > > >
> > > > In the PAG + MT-GRPO experiment, RL is applied after PAG training, but the training objective **only focuses on function call accuracy**. The process does not generate progress-awareness content (and naturally, no loss is computed for the progress-awareness part), resulting in a misalignment of training objectives and a decrease in performance. This highlights that the performance improvement brought by Progra does not solely come from the supervision signal of PAG, but rather from the combination of PAG and the subsequent PAG-RL.
> > > >
> > > > The limitations of PAG training are compensated by the combination of PAG and PAG-RL, with the latter further optimizing the model's multi-turn reasoning and function calling ability through reinforcement learning, leading to a larger overall performance boost. We will continue to conduct further experiments to further validate this point.

---

> > > > > ### Author Response · Authors · 2025-11-25
> > > > >
> > > > > >**W3,W5,Q2,Q4&Q5. Insights of Ablation Experiments, Comparison with More Baselines and Cross-Domain Experiments**
> > > > >
> > > > > Additionally, we have added several additional baselines, including Dr.GRPO and ARTIST, and we also conducted cross-domain experiments on the Telecom subset from Tau2-Bench (all baselines tuned on Airline and Retail and tested on Telecom).
> > > > >
> > > > > New baselines:
> > > > >
> > > > > 1. **ST-GRPO / ST-Dr.GRPO**: We split multi-turn conversation trajectories into single-turn training samples, and then train the model with GRPO or Dr.GRPO, respectively.
> > > > > 2. **MT-GRPO / MT-Dr.GRPO**: We conduct end-to-end reinforcement learning over full multi-turn trajectories as described in the paper, using GRPO and Dr.GRPO as the underlying RL algorithms.
> > > > > 3. **ARTIST**: We adopt the multi-turn RL algorithm proposed in prior work \[1].
> > > > > 4. **SFT+MT-GRPO**: We apply MT-GRPO based on models after SFT.
> > > > >
> > > > > Table. 1:  Performance Comparison on Tau-Bench and Tau2-Bench. "Overall" is average scores of Airline and Retail excluding Telecom.
> > > > > | Backbone LLM | Method | Overall | Airline | Retail | Telecom |
> > > > > |---|---|---|---|---|---|
> > > > > | gpt-4.1 | - | 61.95 | 48.80 | 75.10 | - |
> > > > > | DeepSeek-R1 | - | 58.70 | 53.50 | 63.90 | - |
> > > > > | Gemini 2.5 Pro | - | 62.50 | 48.00 | 77.00 | - |
> > > > > | Qwen2.5-7B | Base Model | 16.60 | 8.00 | 25.20 | 13.62 |
> > > > > | - | SFT | 21.30 | 20.00 | 22.61 | 10.82 |
> > > > > | - | SFT+MT-GRPO | 18.32 | 12.00 | 24.64 | 14.62 |
> > > > > | - | ST-GRPO | 18.19 | 10.00 | 26.38 | 15.94 |
> > > > > | - | ST-Dr.GRPO | 18.33 | 10.00 | 26.67 | 14.49 |
> > > > > | - | MT-GRPO | 19.00 | 11.33 | 26.67 | 12.87 |
> > > > > | - | MT-Dr.GRPO | 18.52 | 10.67 | 26.38 | 12.28 |
> > > > > | - | ARTIST | 18.68 | 8.67  | 28.70 | 15.79 |
> > > > > | - | Progra | **23.91** | **20.00** | **27.83** | **16.23** |
> > > > > | xLAM-2-3B | Base Model | 25.38 | 26.00 | 24.76 | - |
> > > > > | - | SFT | 23.30 | 24.00 | 22.61 | - |
> > > > > | - | SFT+MT-GRPO | 25.54 | 22.67 | 28.41 | - |
> > > > > | - | ST-GRPO | 25.84 | 27.33 | 24.35 | - |
> > > > > | - | ST-Dr.GRPO | 25.81 | 24.67 | 26.96 | - |
> > > > > | - | MT-GRPO | 26.35 | 24.00 | 28.70 | - |
> > > > > | - | MT-Dr.GRPO | 24.32 | 24.00 | 24.64 | - |
> > > > > | - | ARTIST | 25.42 | 25.33 | 25.51 | - |
> > > > > | - | Progra | **31.52** | **30.00** | **33.04** | - |
> > > > > | xLAM-2-8B | Base Model | 46.70 | 35.20 | 58.20 | 10.14 |
> > > > > | - | SFT | 46.57 | 34.00 | 59.13 | 9.65 |
> > > > > | - | SFT+MT-GRPO | 48.96 | 36.00 | 61.92 | 10.82 |
> > > > > | - | ST-GRPO | 48.19 | 36.67 | 59.71 | 10.72 |
> > > > > | - | ST-Dr.GRPO | 50.54 | **39.33** | 61.74 | 10.82 |
> > > > > | - | MT-GRPO | 49.52 | **39.33** | 59.71 | 10.82 |
> > > > > | - | MT-Dr.GRPO | 46.94 | 37.33 | 56.54 | 10.23 |
> > > > > | - | ARTIST | 44.44 | 36.00 | 52.89 | 9.65 |
> > > > > | - | Progra | **51.85** | 38.00 | **65.70**| **13.91** |
> > > > >
> > > > > From the above results, we can conclude that, compared to other baselines such as GRPO, Dr.GRPO, and ARTIST, Progra shows **substantial improvements** on most tasks. Additionally, under cross-domain evaluation (training on the airline and retail domains and testing on the telecom domain), Progra continues to demonstrate significant advantages across multiple baseline models. Although some baselines (such as ST-GRPO and MT-GRPO) also show improvements, Progra consistently outperforms these methods, highlighting its unique strengths in **in-domain and out-domain tasks**. These results further support the effectiveness of our approach, demonstrating that Progra not only adapts well to specific task domains but also maintains strong generalization across different domains.
> > > > >
> > > > > Additionally, RLFactory belongs to the RL infrastructure work and is not part of the RL tunning algorithms, so we do not recommend it as a baseline for comparison. We will conduct more experiments on additional benchmarks and baselines (such as SPA-RL) in the upcoming period.
> > > > >
> > > > >
> > > > > \[1] Prabhakar, A., Liu, Z., Zhu, M., Zhang, J., Awalgaonkar, T., Wang, S., ... & Xiong, C. Apigen-mt: Agentic pipeline for multi-turn data generation via simulated agent-human interplay, 2025. URL https://arxiv.org/abs/2504.03601.
> > > > >
> > > > > \[2] Singh, J., Magazine, R., Pandya, Y., & Nambi, A. (2025). Agentic reasoning and tool integration for llms via reinforcement learning. arXiv preprint arXiv:2505.01441.

---

> > > > > > ### Author Response · Authors · 2025-11-25
> > > > > >
> > > > > > >**W4&Q3. Efficiency and Performance Trade-offs in Progra**
> > > > > >
> > > > > > Your comment on efficiency is highly important. During training, each action in Progra is decomposed into two sub-steps, (i) explicitly generating progress-awareness content and (ii) issuing the function call, which inevitably introduces additional time and token overhead. To quantify this, we measure the per-step GPU time and the total number of generated tokens for xLAM-2-3B when training on Tau-Bench. The results are reported below:
> > > > > >
> > > > > > Table. 2:  Efficiency Comparison on Tau-Bench.
> > > > > > | Backbone LLM | Method | Score | Improv. | Overall Traing Time (GPU*h) | Time/Step (GPU*s) | #Token/Step |
> > > > > > |---|---|---|---|---|---|---|
> > > > > > | xLAM2-3B | Base Model | 25.45 | - | - | - | - |
> > > > > > | - | MT-GRPO | 26.35 | 3.53 | ～260 | 1903.20 | 1040.67 |
> > > > > > | - | Progra | 31.52 | 23.85 |~120 | 2900.16 | 2243.32 |
> > > > > >
> > > > > >
> > > > > > From the table, Progra requires approximately **50\%** more training time per step than MT-GRPO, while achieving about a **19.62\%** higher overall performance on Tau-Bench. Therefore, the overall cost–benefit trade-off remains well balanced. Moreover, MT-GRPO improves over the base model by only **3.53\%**, whereas Progra yields a **23.85\%** improvement over the base model, indicating a substantially larger relative gain. This demonstrates that the additional time and token costs during training deliver a considerable return in performance.
> > > > > >
> > > > > > At test time, Progra and all baselines use the default prompts provided by BFCL and Tau-Bench. The model outputs only explicit reasoning content and function calls, rather than explicitly generating long progress-awareness sections. Instead, progress-awareness is implicitly reflected within the reasoning outputs, ensuring fair comparison without introducing artificial inference-time scaling. This design preserves practical deployability and avoids inflating evaluation performance through increased test-time computation.
> > > > > >
> > > > > >
> > > > > > >**W6. Expanding Evaluation and Addressing Case Study Insights**
> > > > > >
> > > > > > Thank you for raising this concern. Due to resource constraints, we regret that we were unable to conduct experiments on additional benchmarks during the first week. However, we will make every effort to update the results on more benchmarks during the remaining time of the rebuttal. Specifically, we are considering evaluating on more complex benchmarks like ComplexFuncBench to further assess the robustness of our method. If you have any suggestions for specific benchmarks or additional experiments that you think would be valuable, we would greatly appreciate your input.
> > > > > >
> > > > > >
> > > > > > Additionally, Table 3 in Section 4.6 (Case Study) in the paper presents the results from our experiments on the Tau-Bench Airline subset. The statistical results indicate that, in this subset, Progra requires fewer average response steps than the base model, demonstrating an improvement in function call efficiency brought by Progra. We believe this result helps address the misunderstanding regarding the cherry-picked cases.
> > > > > >
> > > > > >
> > > > > > We appreciate the reviewer’s feedback and will make the necessary improvements in the revised version.

---

### Official Review · Reviewer_5HHV · 2025-10-31

**Soundness:** 2
**Presentation:** 3
**Contribution:** 2
**Rating:** 4
**Confidence:** 3

**Summary:**

This work introduces Progra, a framework that explicitly incorporates progress awareness into LLM training for multi-turn function calling. Progra combines (i) a Progress Awareness Generation (PAG) pipeline, which automatically constructs datasets coupling conversation summaries with future task planning, and (ii) a Progress Awareness-Guided Reinforcement Learning (PAG-RL) algorithm, which integrates progress awareness into RL training to reduce contextual redundancy and improve alignment between local actions and global task completion. Empirical results on two public benchmarks demonstrate that Progra significantly outperforms existing methods, highlighting the effectiveness of progress awareness in enabling robust and efficient multi-turn function calling.

**Strengths:**

This work makes substantive contributions to multi-turn function calling through the explicit formalization and integration of progress awareness into LLM training. The originality lies in identifying progress awareness as a compositional capability encompassing both retrospective summarization and prospective planning, operationalized through a principled two-phase framework that couples synthetic data generation with awareness-guided reinforcement learning. The model-aware verification mechanism represents a methodologically sound approach to quality control, leveraging the target policy itself as an oracle for data validation. The technical execution demonstrates rigor through formal MDP-based problem formulation, comprehensive empirical evaluation spanning multiple benchmarks and models, and systematic ablation studies that establish the necessity of both framework components. The experimental design effectively isolates the contribution of progress awareness through controlled comparisons against baselines. The work exhibits strong clarity in its presentation, with coherent mathematical notation.

**Weaknesses:**

1. Only evaluation on two benchmarks (BFCL, τ-Bench). Can you provide more results on other benchmarks?

2. PAG-RL requires generating awareness text before each action, at least doubling inference cost. Section A.1 acknowledges additional time costs but provides zero quantitative analysis. This is critical for practical deployment and represents a major gap.

3. Only compare against vanilla MT-GRPO for RL. Can you add comparisons with SPA-RL or other methods? Or demonstrate that MT-GRPO is a strong baseline by comparing it against these methods?

4. Based on the ablation study in Table 2, there exists a substantial performance discrepancy between Qwen2.5-7B and xLAM models suggesting the method maybe lacks robustness.

**Questions:**

Please  refer to the Weaknesses.

---

> ### Author Response · Authors · 2025-11-25
>
> >**W1. Expansion of Benchmarks**
>
> Thank you for pointing out this limitation. Due to resource constraints, we regret that we were unable to conduct experiments on additional benchmarks during the first week. However, we will make every effort to update the results on more benchmarks during the remaining time of the rebuttal. Specifically, we are considering evaluating on more complex benchmarks like ComplexFuncBench to further assess the robustness of our method. If you have any suggestions for specific benchmarks or additional experiments that you think would be valuable, we would greatly appreciate your input.
>
> >**W2. Insights of Efficiency and Performance Trade-offs in Progra**
>
> Your comment on efficiency trade-offs is highly important. During training, each action in Progra is decomposed into two sub-steps, (i) explicitly generating progress-awareness content and (ii) issuing the function call, which inevitably introduces additional time and token overhead. To quantify this, we measure the per-step GPU time and the total number of generated tokens for xLAM-2-3B when training on Tau-Bench. The results are reported below:
>
> Table. 1:  Efficiency Comparison on Tau-Bench. "Score" are average scores on Tau-Bench.
> | Backbone LLM | Method | Score | Improv.(%) | Time/Step (GPU*s) | #Token/Step |
> |---|---|---|---|---|---|
> | xLAM-2-3B | Base Model | 25.45  | - | - | - |
> | - | MT-GRPO | 26.35 | 3.53 | 1903.20 | 1040.67 |
> | - | Progra | 31.52 | 23.85 | 2900.16 | 2243.32 |
>
> From the table, Progra requires approximately **50\%** more training time per step than MT-GRPO, while achieving about a **19.62\%** higher overall performance on Tau-Bench. Therefore, the overall cost–benefit trade-off remains well balanced. Moreover, MT-GRPO improves over the base model by only **3.53\%**, whereas Progra yields a **23.85\%** improvement over the base model, indicating a substantially larger relative gain. This demonstrates that the additional time and token costs during training deliver a considerable return in performance.
>
> At test time, Progra and all baselines use the default prompts provided by BFCL and Tau-Bench. The model outputs only explicit reasoning content and function calls, rather than explicitly generating long progress-awareness sections. Instead, progress-awareness is implicitly reflected within the reasoning outputs, ensuring fair comparison without introducing artificial inference-time scaling. This design preserves practical deployability and avoids inflating evaluation performance through increased test-time computation.

---

> > ### Author Response · Authors · 2025-11-25
> >
> > >**W3. Comparison With Enhanced Baselines**
> >
> > Thank you for raising this concern. We have added several additional baselines, including Dr.GRPO and ARTIST:
> >
> > 1. **ST-GRPO / ST-Dr.GRPO**: We split multi-turn conversation trajectories into single-turn training samples, and then train the model with GRPO or Dr.GRPO, respectively.
> > 2. **MT-GRPO / MT-Dr.GRPO**: We conduct end-to-end reinforcement learning over full multi-turn trajectories as described in the paper, using GRPO and Dr.GRPO as the underlying RL algorithms.
> > 3. **ARTIST**: We adopt the multi-turn RL algorithm proposed in prior work \[1].
> > 4. **SFT+MT-GRPO**: We apply MT-GRPO based on models after SFT.
> >
> > Table. 1:  Performance Comparison on Tau-Bench. "Overall" is average scores of Airline and Retail.
> > | Backbone LLM | Method | Overall | Airline | Retail |
> > |---|---|---|---|---|
> > | gpt-4.1 | - | 61.95 | 48.80 | 75.10 |
> > | DeepSeek-R1 | - | 58.70 | 53.50 | 63.90 |
> > | Gemini 2.5 Pro | - | 62.50 | 48.00 | 77.00 |
> > | Qwen2.5-7B | Base Model | 16.60 | 8.00 | 25.20 |
> > | - | SFT | 21.30 | 20.00 | 22.61 |
> > | - | SFT+MT-GRPO | 18.32 | 12.00 | 24.64 |
> > | - | ST-GRPO | 18.19 | 10.00 | 26.38 |
> > | - | ST-Dr.GRPO | 18.33 | 10.00 | 26.67 |
> > | - | MT-GRPO | 19.00 | 11.33 | 26.67 |
> > | - | MT-Dr.GRPO | 18.52 | 10.67 | 26.38 |
> > | - | ARTIST | 18.68 | 8.67  | 28.70 |
> > | - | Progra | **23.91** | **20.00** | **27.83** |
> > | xLAM-2-3B | Base Model | 25.38 | 26.00 | 24.76 |
> > | - | SFT | 23.30 | 24.00 | 22.61 |
> > | - | SFT+MT-GRPO | 25.54 | 22.67 | 28.41 |
> > | - | ST-GRPO | 25.84 | 27.33 | 24.35 |
> > | - | ST-Dr.GRPO | 25.81 | 24.67 | 26.96 |
> > | - | MT-GRPO | 26.35 | 24.00 | 28.70 |
> > | - | MT-Dr.GRPO | 24.32 | 24.00 | 24.64 |
> > | - | ARTIST | 25.42 | 25.33 | 25.51 |
> > | - | Progra | **31.52** | **30.00** | **33.04** |
> > | xLAM-2-8B | Base Model | 46.70 | 35.20 | 58.20 |
> > | - | SFT | 46.57 | 34.00 | 59.13 |
> > | - | SFT+MT-GRPO | 48.96 | 36.00 | 61.92 |
> > | - | ST-GRPO | 48.19 | 36.67 | 59.71 |
> > | - | ST-Dr.GRPO | 50.54 | **39.33** | 61.74 |
> > | - | MT-GRPO | 49.52 | **39.33** | 59.71 |
> > | - | MT-Dr.GRPO | 46.94 | 37.33 | 56.54 |
> > | - | ARTIST | 44.44 | 36.00 | 52.89 |
> > | - | Progra | **51.85** | 38.00 | **65.70**|
> >
> > From the above results, we can conclude that, compared to other baselines such as GRPO, Dr.GRPO, and ARTIST, Progra shows substantial improvements on most tasks. continues to demonstrate significant advantages across multiple baseline models.
> > These results further support the effectiveness of our approach, demonstrating that Progra adapts well to specific task domains.
> >
> > We will provide more comprehensive experimental results in the updated paper PDF as soon as possible.
> >
> > \[1] Singh, J., Magazine, R., Pandya, Y., & Nambi, A. (2025). Agentic reasoning and tool integration for llms via reinforcement learning. arXiv preprint arXiv:2505.01441.
> >
> >
> > >**W4. Robustness Across Models**
> >
> > Thank you for your valuable question. Based on the ablation study in Table 2 in our paper, there indeed exists a substantial performance discrepancy between Qwen2.5-7B and the xLAM models. However, we believe that this discrepancy does not indicate a lack of robustness in the method, for the following reasons:
> >
> > First, Qwen2.5-7B, xLAM-2-3B, and xLAM-2-8B are commonly used models in function calling domain **with different model sizes and architectures**. Specifically, xLAM-2-3B is based on the Qwen2.5 architecture, while xLAM-2-8B is based on the Llama architecture. The differences of architectural designs and training approaches makes it expected that they would perform differently on function calling tasks.
> >
> > Second, Qwen2.5-7B is a general-purpose model, whereas the xLAM-2 series is **specifically optimized for function calling tasks** and has been trained according to the methods outlined in the APIGEN-MT paper \[2]. Therefore, it is natural for xLAM-2 to outperform Qwen2.5-7B on function calling tasks.
> >
> > Third, despite the differences in architecture and task adaptability, our Progra method has shown **significant performance improvements** across various model sizes and use cases. This further demonstrates the applicability and robustness of the Progra method, especially as it yields positive results across models of different sizes.
> >
> > To further validate the robustness of the Progra method, we will provide additional experiments with various backbone LLMs during the discussion phase, which will further expand and enrich our understanding of model adaptability.
> >
> > \[2] Prabhakar, A., Liu, Z., Zhu, M., Zhang, J., Awalgaonkar, T., Wang, S., ... & Xiong, C. Apigen-mt: Agentic pipeline for multi-turn data generation via simulated agent-human interplay, 2025. URL https://arxiv.org/abs/2504.03601.
> >
> > We sincerely appreciate the reviewer’s thoughtful and valuable comments and we are hoping for more discussion from the reviewer.

---

### Official Review · Reviewer_2Gcy · 2025-11-01

**Soundness:** 2
**Presentation:** 2
**Contribution:** 2
**Rating:** 4
**Confidence:** 3

**Summary:**

PROGRA is a two-phase training framework that enhances large language models’ performance on multi-turn function calling by explicitly incorporating progress awareness---the ability to track task progress, summarize relevant history, and plan subsequent actions.

Phase 1 (Progress Awareness Generation, PAG) constructs synthetic training data that couples conversation histories with automatically generated progress summaries and future plans. A strong generator LLM produces these summaries, which are verified by testing whether they contain sufficient information to reconstruct the correct function calls, augmented for diversity, and then used to warm up the target model through supervised fine-tuning.

Phase 2 (Progress Awareness–Guided Reinforcement Learning, PAG-RL) introduces progress awareness into the rollout process: before each function call, the model generates a concise awareness summary and conditions its next action on it. The combined behavior--summary and action---is optimized end-to-end with the GRPO (Group Relative Policy Optimization) algorithm using a composite reward that balances structural validity, functional correctness, task success, and efficiency.

Evaluated on BFCL-V3 and τ-Bench benchmarks using Qwen2.5-7B, xLAM-2-3B, and xLAM-2-8B backbones (200 examples per benchmark), PROGRA consistently outperforms baselines including supervised fine-tuning and vanilla multi-turn GRPO. Reported relative gains range from roughly 5 to 40% depending on model and dataset, with the best overall accuracy reaching ≈52%. The results demonstrate that explicitly modeling progress awareness improves coherence, reduces redundant calls, and strengthens long-horizon task execution, though at the cost of additional computation during training.

**Strengths:**

### 1. **Well-Designed Data Synthesis Pipeline with Quality Verification**

The Progress Awareness Generation (PAG) pipeline includes a genuinely clever verification step: after generating progress summaries using a strong LLM, it tests whether these summaries contain sufficient information by attempting to reconstruct the correct function call from the summary alone, keeping only summaries that pass this test. This "model-aware verification" ensures the synthetic training data is actually useful for the target model rather than just plausible-sounding text. Combined with diversity-preserving augmentation strategies (paraphrasing, schema perturbations, word masking), this creates a principled approach to synthetic data generation that could be adapted to other domains. The pipeline is fully automated and appears reproducible based on the provided implementation details.

### 2. **Consistent Improvements with Proper Ablations**

Despite varying magnitude, PROGRA shows improvements over baselines across all tested configurations (three model sizes, two benchmarks, multiple evaluation subsets), with the full method consistently outperforming partial implementations. The ablation studies systematically demonstrate that both phases contribute to performance, and the paper provides sufficient implementation details (training procedures, hyperparameters, reward functions) for reproduction. The gains on τ-Bench are substantial for smaller models (44% relative improvement for Qwen2.5-7B), and even the more modest improvements on other settings suggest the approach provides real value rather than being entirely spurious. The case study showing both higher accuracy (26%→40%) and fewer steps (27.08→23.30) provides supporting evidence that the method produces more efficient and coherent trajectories.

**Weaknesses:**

### 1. **Insufficient Evaluation of Efficiency Trade-offs**

The method generates substantially more tokens per turn (estimated ~320 tokens vs ~20 tokens for direct function calling due to `<summary>` and `<think>` sections), likely resulting in ~15x higher output token costs and proportionally increased latency. While the paper shows fewer function calls (23.30 vs 27.08 average steps), which could save on API costs, it provides no analysis of the overall cost-benefit trade-off. Without reporting token counts, latency measurements, or cost modeling, it's impossible to assess whether the accuracy improvements (which range from 1-44% relative gain) justify the computational overhead, or to determine under what conditions (expensive APIs vs. cheap APIs, latency-sensitive vs. batch processing) this approach would be practical to deploy.

### 2. **Weak Baselines and Limited Generalization Testing**

The evaluation compares only against basic baselines (base model, simple prompting, standard SFT, vanilla GRPO) and lacks comparison to stronger alternatives like other structured reasoning approaches, advanced prompt engineering, or state-of-the-art commercial models. More critically, all models are trained and tested on the same domain (BFCL→BFCL, τ-Bench→τ-Bench) with small datasets (200 examples each), raising concerns about overfitting. The paper provides no cross-domain evaluation (train on airline, test on retail), no out-of-distribution testing, and no analysis of how performance scales with more training data. Combined with inconsistent results across settings (1% gains on some benchmarks, 44% on others, with ablations showing contradictory effects across models) and lack of statistical significance testing (only 3 trials), it's unclear whether the improvements represent genuine capability gains or artifacts of the specific evaluation setup.

**Questions:**

Q1. The ablations show interesting variance - Phase 1 alone hurts performance on Qwen/τ-Bench but helps on xLAM-2-8B. Have you tested cross-domain generalization (e.g., train on airline, test on retail, or train on BFCL, test on τ-Bench)? What do you think explains the different component contributions across model sizes?

Q2. Since PROGRA introduces explicit <summary> and <think> generations at each turn, have you measured the additional token and latency overhead relative to vanilla GRPO? It would be helpful to understand whether the observed accuracy gains translate into overall efficiency improvements, and in what deployment settings you think this trade-off is worthwhile.

---

> ### Author Response · Authors · 2025-11-25
>
> >**W1&Q2. Insights of Efficiency and Performance Trade-offs in Progra**
>
> Your comment on efficiency trade-offs is highly important. During training, each action in Progra is decomposed into two sub-steps, (i) explicitly generating progress-awareness content and (ii) issuing the function call, which inevitably introduces additional time and token overhead. To quantify this, we measure the per-step GPU time and the total number of generated tokens for xLAM-2-3B when training on Tau-Bench. The results are reported below:
>
> Table. 1:  Efficiency Comparison on Tau-Bench. "Score" are average scores on Tau-Bench.
> | Backbone LLM | Method | Score | Improv.(%) | Time/Step (GPU*s) | #Token/Step |
> |---|---|---|---|---|---|
> | xLAM-2-3B | Base Model | 25.45  | - | - | - |
> | - | MT-GRPO | 26.35 | 3.53 | 1903.20 | 1040.67 |
> | - | Progra | 31.52 | 23.85 | 2900.16 | 2243.32 |
>
> From the table, Progra requires approximately **50\%** more training time per step than MT-GRPO, while achieving about a **19.62\%** higher overall performance on Tau-Bench. Therefore, the overall cost–benefit trade-off remains well balanced. Moreover, MT-GRPO improves over the base model by only **3.53\%**, whereas Progra yields a **23.85\%** improvement over the base model, indicating a substantially larger relative gain. This demonstrates that the additional time and token costs during training deliver a considerable return in performance.
>
> At test time, Progra and all baselines use the default prompts provided by BFCL and Tau-Bench. The model outputs only explicit reasoning content and function calls, rather than explicitly generating long progress-awareness sections. Instead, progress-awareness is implicitly reflected within the reasoning outputs, ensuring fair comparison without introducing artificial inference-time scaling. This design preserves practical deployability and avoids inflating evaluation performance through increased test-time computation.

---

> > ### Author Response · Authors · 2025-11-25
> >
> > >**W2. Comparison With Enhanced Baselines and Cross-Domain Experiments**
> >
> > Thank you for raising this concern. We have added several additional baselines, including Dr.GRPO and ARTIST, and we also conducted cross-domain experiments on the Telecom subset from Tau2-Bench (all baselines trained on Airline and Retail and tested on Telecom).
> >
> > New baselines:
> >
> > 1. **ST-GRPO / ST-Dr.GRPO**: We split multi-turn conversation trajectories into single-turn training samples, and then train the model with GRPO or Dr.GRPO, respectively.
> > 2. **MT-GRPO / MT-Dr.GRPO**: We conduct end-to-end reinforcement learning over full multi-turn trajectories as described in the paper, using GRPO and Dr.GRPO as the underlying RL algorithms.
> > 3. **ARTIST**: We adopt the multi-turn RL algorithm proposed in prior work \[1].
> > 4. **SFT+MT-GRPO**: We apply MT-GRPO based on models after SFT.
> >
> > Table. 2:  Performance Comparison on Tau-Bench and Tau2-Bench. "Overall" is average scores of Airline and Retail excluding Telecom.
> > | Backbone LLM | Method | Overall | Airline | Retail | Telecom |
> > |---|---|---|---|---|---|
> > | gpt-4.1 | - | 61.95 | 48.80 | 75.10 | - |
> > | DeepSeek-R1 | - | 58.70 | 53.50 | 63.90 | - |
> > | Gemini 2.5 Pro | - | 62.50 | 48.00 | 77.00 | - |
> > | Qwen2.5-7B | Base Model | 16.60 | 8.00 | 25.20 | 13.62 |
> > | - | SFT | 21.30 | 20.00 | 22.61 | 10.82 |
> > | - | SFT+MT-GRPO | 18.32 | 12.00 | 24.64 | 14.62 |
> > | - | ST-GRPO | 18.19 | 10.00 | 26.38 | 15.94 |
> > | - | ST-Dr.GRPO | 18.33 | 10.00 | 26.67 | 14.49 |
> > | - | MT-GRPO | 19.00 | 11.33 | 26.67 | 12.87 |
> > | - | MT-Dr.GRPO | 18.52 | 10.67 | 26.38 | 12.28 |
> > | - | ARTIST | 18.68 | 8.67  | 28.70 | 15.79 |
> > | - | Progra | **23.91** | **20.00** | **27.83** | **16.23** |
> > | xLAM-2-3B | Base Model | 25.38 | 26.00 | 24.76 | - |
> > | - | SFT | 23.30 | 24.00 | 22.61 | - |
> > | - | SFT+MT-GRPO | 25.54 | 22.67 | 28.41 | - |
> > | - | ST-GRPO | 25.84 | 27.33 | 24.35 | - |
> > | - | ST-Dr.GRPO | 25.81 | 24.67 | 26.96 | - |
> > | - | MT-GRPO | 26.35 | 24.00 | 28.70 | - |
> > | - | MT-Dr.GRPO | 24.32 | 24.00 | 24.64 | - |
> > | - | ARTIST | 25.42 | 25.33 | 25.51 | - |
> > | - | Progra | **31.52** | **30.00** | **33.04** | - |
> > | xLAM-2-8B | Base Model | 46.70 | 35.20 | 58.20 | 10.14 |
> > | - | SFT | 46.57 | 34.00 | 59.13 | 9.65 |
> > | - | SFT+MT-GRPO | 48.96 | 36.00 | 61.92 | 10.82 |
> > | - | ST-GRPO | 48.19 | 36.67 | 59.71 | 10.72 |
> > | - | ST-Dr.GRPO | 50.54 | **39.33** | 61.74 | 10.82 |
> > | - | MT-GRPO | 49.52 | **39.33** | 59.71 | 10.82 |
> > | - | MT-Dr.GRPO | 46.94 | 37.33 | 56.54 | 10.23 |
> > | - | ARTIST | 44.44 | 36.00 | 52.89 | 9.65 |
> > | - | Progra | **51.85** | 38.00 | **65.70**| **13.91** |
> >
> > From the above results, we can conclude that, compared to other baselines such as GRPO, Dr.GRPO, and ARTIST, Progra shows **substantial improvements** on most settings and tasks. Additionally, under cross-domain evaluation (training on the airline and retail domains and testing on the telecom domain), Progra continues to demonstrate significant advantages across multiple baseline models. Although some baselines (such as ST-GRPO and MT-GRPO) also show improvements, Progra consistently outperforms these methods, highlighting its unique strengths in **in-domain and out-domain performances**. These results further support the effectiveness of our approach, demonstrating that Progra not only adapts well to specific task domains but also maintains strong generalization across different domains.
> >
> > We will update the scalability experiments regarding the increase in training data in the near future. Additionally, we will provide more comprehensive experimental results in the updated paper PDF.
> >
> >
> > \[1] Singh, J., Magazine, R., Pandya, Y., & Nambi, A. (2025). Agentic reasoning and tool integration for llms via reinforcement learning. arXiv preprint arXiv:2505.01441.

---

> > > ### Author Response · Authors · 2025-11-25
> > >
> > > >**Q1. Ablation Insights**
> > >
> > > Thanks very much for raising this question. We conducted cross-domain experiments by training on the Airline and Tetail domains and testing on the Telecom domain (from Tau2-Bench), demonstrating that Progra consistently outperforms other baselines, highlighting its **strong generalization** across both in-domain and out-of-domain tasks. For details of the cross-domain experiments, please refer to our response to Weakness 2 above.
> > >
> > > In the ablation study results presented in Table 2 in the paper, the gain from "w/o PAG-RL" (i.e., performing only the first stage of PAG training) is generally smaller compared to the gains from other components. We believe this is due to a **misalignment between the training and evaluation objectives**. During PAG training, we focus on enhancing the LLM's ability to summarize the current dialogue progress and plan for future steps, but do not directly enhance its multi-turn function calling capability. This subtle misalignment in training objectives leads to relatively smaller performance improvements and may occasionally result in degraded performance (such as the Qwen/Tau-Bench data point you mentioned). This indicates that the performance improvement brought by Progra does not solely come from the supervision signal of PAG, but rather from the combination of PAG and the subsequent PAG-RL.
> > >
> > > The PAG training alone is insufficient, and this gap is **addressed by the combination of PAG and PAG-RL**, with the latter further optimizing the model’s multi-turn reasoning and function calling ability via reinforcement learning, leading to a larger overall performance boost.
> > >
> > > Regarding the contribution of different model sizes, we speculate that smaller models, such as xLAM-2-3B, may benefit more from the PAG training phase, while larger models (xLAM-2-8B, Qwen2.5-7B) may require further reinforcement learning optimization from PAG-RL to fully realize their potential. We will continue to conduct further experiments to validate this hypothesis.
> > >
> > >
> > > We thank the reviewer for carefully pointing out these issues and we hope for further discussions.

---

### Official Review · Reviewer_odGe · 2025-11-05

**Soundness:** 2
**Presentation:** 2
**Contribution:** 2
**Rating:** 4
**Confidence:** 4

**Summary:**

Most large language model training with reinforcement learning typically solves problems using single-step reinforcement learning, ignoring the fact that models interact with people over multiple turns. Specifically, the dialogue leading up to the final turn is treated as context, and the model is only trained to predict the last turn in the dialogue. The authors propose to address this issue by introducing a framework called Progra, which explicitly incorporates progress awareness into LLM training for multi-turn function calling. Progra combines two key components: Progress Awareness Generation (PAG) and Progress Awareness Guided Reinforcement Learning (PAG-RL). PAG is designed to generate high-quality datasets consisting of metadata that includes tool descriptions and conversation contexts—essentially, the history of the conversation. Meanwhile, PAG-RL learns how to sample responses based on summaries of this PAG history. The authors conduct several ablation studies, comparing their framework against models that use either PAG, PAG-RL, or both. They demonstrate that their proposed framework outperforms existing methods across two benchmarks, BFCL and \tau-Bench.

**Strengths:**

- The ability to train LLMs for multi-turn conversations is a crucial problem.
- The authors proposed a novel strategy to enable LLMs not only to learn in a multi-turn setting but also to capture information in a way that allows them to build on previous utterances.
- The description of the proposed method was clear, and I was able to understand the significance of the proposed approach.

**Weaknesses:**

- The baseline algorithms, such as GRPO, are not being compared fairly. Specifically, PAG is essentially another form of supervised fine-tuning, whereas MT-GRPO is not fine-tuned in a supervised manner at all. Combining PAG with GRPO does not count because PAG is a different format that includes information and metadata, which GRPO cannot utilize effectively.
- The results do not present standard deviations, making it unclear how significant the findings are.

**Questions:**

- If you supervised fine-tuning of the baseline with in-domain data, how well does it perform? Additionally, if you applied GRPO on top of this newly supervised fine-tuned baseline, what results did you achieve?

- What does "w/o PAG" mean in the plots? Does it indicate that you are running PAG-RL on top of the base model? If so, how does the baseline, which does not understand how to emit compatible progress awareness information (meta), achieve this without training first?

- GRPO is known to have subtle issues in optimization. Did you consider other variants, such as Dr.GRPO, which address some of the shortcomings of GRPO?

- Did you perform any qualitative analysis on why the proposed multi-turn framework performs better? Specifically, is there anything in the utterances generated by the model that suggests why it is performing better?

- Did you try training with different prompts using MT-GRPO on the benchmark tasks evaluated in the paper?

---

> ### Author Response · Authors · 2025-11-25
>
> >**W1&Q1. Fair Comparison with Enhanced Baselines**
>
> Thank you very much for raising this concern. To ensure a fair comparison with the baselines and to avoid any potential advantage introduced by additional supervision signals during Progra training, we report a stronger and more comparable baseline as follows: **based on the same SFT-trained model, we further conduct GRPO training (i.e., SFT + GRPO) while excluding any progress-awareness-related information.** This baseline is able to fully leverage the benefits of supervised fine-tuning and is aligned with the RL training setup, yet it does not access the progress-aware signals that are specific to Progra.
>
> We conduct experiments on Tau-Bench, and the results are summarized in Table 1. The results demonstrate that in-domain SFT fine-tuning improves model performance, and conducting multiple additional rounds of GRPO on top of this SFT initialization **further enhances performance**. And as can be observed, the results demonstrate that under this fair setting, where the baseline enjoys the same SFT initialization and subsequent GRPO optimization but without progress-aware cues, **Progra consistently outperforms the corresponding enhanced baselines** as well as the base model across all three backbone LLMs. This indicates that the gains of Progra are not attributable to extra supervision or to information packaged in a format unusable by GRPO; rather, they stem from the proposed progress-aware reinforcement learning mechanism itself. Therefore, the observed improvements reliably support the effectiveness of our method.
>
> Table. 1:  Performance Comparison between SFT / SFT+GRPO and Progra on Tau-Bench.
> | Backbone LLM | Method | Overall | Airline | Retail |
> |---|---|---|---|---|
> | Qwen2.5-7B | Base Model | 16.60 | 8.00 | 25.20 |
> | - | SFT | 21.30 | **20.00** | 22.61 |
> | - | SFT+MT-GRPO | 18.32 | 12.00 | 24.64 |
> | - | Progra | **23.91** | **20.00** | **27.83** |
> | xLAM-2-3B | Base Model | 25.38 | 26.00 | 24.76 |
> | - | SFT | 23.30 | 24.00 | 22.61 |
> | - | SFT+MT-GRPO |  25.54 | 22.67 | 28.41 |
> | - | Progra | **31.52** | **30.00** | **33.04** |
> | xLAM-2-8B | Base Model | 46.70 | 35.20 | 58.20 |
> | - | SFT | 46.57 | 34.00 | 59.13 |
> | - | SFT+MT-GRPO |  48.96 | 36.00 | 61.92 |
> | - | Progra | **51.85** | **38.00** | **65.70**|
>
> We will provide more complete experimental results in the updated paper PDF soon.
>
>
> >**W2. Inclusion of Statistical Significance**
>
> Thank you very much for pointing out this omission. We will report the experimental results together with their standard deviations in the updated paper PDF as soon as possible.
>
>
>
> >**Q2. Clarification on Progress Awareness**
>
> "w/o PAG" indeed denotes running PAG-RL on top of the base model. In our experiments, both general-purpose models such as Qwen2.5 and domain-specialized models such as xLAM-2-8b exhibit a basic ability for progress-aware generation under prompt guidance. Figure 4 in the paper shows that an untrained Qwen2.5-7B, relying solely on its own progress-awareness inferred from the current conversation, achieves an accuracy of approximately 17% on predicting the next function call (on validation samples).

---

> > ### Author Response · Authors · 2025-11-25
> >
> > >**Q3. Comparison With Enhanced Baselines**
> >
> > Thanks for raising this question. we further include the following baselines and commercial LLMs:
> >
> > 1. **ST-GRPO / ST-Dr.GRPO:** We split multi-turn conversation trajectories into single-turn training samples, and then train the model with GRPO or Dr.GRPO, respectively.
> > 2. **MT-GRPO / MT-Dr.GRPO:** We conduct end-to-end reinforcement learning over full multi-turn trajectories as described in the paper, using GRPO and Dr.GRPO as the underlying RL algorithms.
> > 3. **ARTIST:** We adopt the multi-turn RL algorithm proposed in prior work \[1].
> >
> > The experimental results are reported in the table below. As shown, across all three backbone LLMs, Progra achieves the best overall performance among all baselines, which provides strong evidence for the effectiveness of our method.
> >
> >
> > Table. 2:  Performance Comparison on Tau-Bench.
> > | Backbone LLM | Method | Overall | Airline | Retail |
> > |---|---|---|---|---|
> > | gpt-4.1 | - | 61.95 | 48.80 | 75.10 |
> > | DeepSeek-R1 | - | 58.70 | 53.50 | 63.90 |
> > | Gemini 2.5 Pro | - | 62.50 | 48.00 | 77.00 |
> > | Qwen2.5-7B | Base Model | 16.60 | 8.00 | 25.20 |
> > | - | ST-GRPO | 18.19 | 10.00 | 26.38 |
> > | - | ST-Dr.GRPO | 18.33 | 10.00 | 26.67 |
> > | - | MT-GRPO | 19.00 | 11.33 | 26.67 |
> > | - | MT-Dr.GRPO | 18.52 | 10.67 | 26.38 |
> > | - | ARTIST | 18.68 | 8.67  | 28.70 |
> > | - | Progra | **23.91** | **20.00** | **27.83** |
> > | xLAM-2-3B | Base Model | 25.38 | 26.00 | 24.76 |
> > | - | ST-GRPO | 25.84 | 27.33 | 24.35 |
> > | - | ST-Dr.GRPO | 25.81 | 24.67 | 26.96 |
> > | - | MT-GRPO | 26.35 | 24.00 | 28.70 |
> > | - | MT-Dr.GRPO | 24.32 | 24.00 | 24.64 |
> > | - | ARTIST | 25.42 | 25.33 | 25.51 |
> > | - | Progra | **31.52** | **30.00** | **33.04** |
> > | xLAM-2-8B | Base Model | 46.70 | 35.20 | 58.20 |
> > | - | ST-GRPO | 48.19 | 36.67 | 59.71 |
> > | - | ST-Dr.GRPO | 50.54 | **39.33** | 61.74 |
> > | - | MT-GRPO | 49.52 | **39.33** | 59.71 |
> > | - | MT-Dr.GRPO | 46.94 | 37.33 | 56.54 |
> > | - | ARTIST | 44.44 | 36.00 | 52.89 |
> > | - | Progra | **51.85** | 38.00 | **65.70**|
> >
> > \[1] Singh, J., Magazine, R., Pandya, Y., & Nambi, A. (2025). Agentic reasoning and tool integration for llms via reinforcement learning. arXiv preprint arXiv:2505.01441.
> >
> >
> > >**Q4. Insights into Performance of Progress Awareness**
> >
> > Thank you for this question, which allows us to further clarify our insights.
> >
> > First, at a high level, explicit progress awareness makes the model’s decisions (i.e., function calls) more **globally consistent**. During training, before taking each action, Progra generates an explicit summary and plan, which more clearly aligns the next action with the overall task objective, rather than relying solely on information from the current turn.
> >
> > Second, in multi-turn function-calling tasks, conversations are typically long, and LLMs may overlook **important information from earlier turns**. Progra’s progress-awareness component encourages the model to revisit previously confirmed information (e.g., passenger details, existing reservation status, and outcomes of prior successful/failed calls) and to propose the minimal necessary next-step plan accordingly. This reduces common failure modes such as repeatedly invoking the same function or backtracking to fill in missing critical arguments.
> >
> >
> > >**Q5. Exploring Prompt Strategies for Better Performance**
> >
> > During training for MT-GRPO and Progra, we experimented with multiple prompt strategies and loss calculations, including the following:
> >
> > **MT-GRPO:**
> >
> > 1. Directly generating function calls.
> > 2. Jointly generating explicit reasoning content and function calls in one response.
> > 3. Generating explicit reasoning content and function calls in two steps, while masking out the loss on the reasoning content.
> > 4. Generating explicit reasoning content and function calls in two steps, and including both parts in the loss calculation.
> >
> > **Progra:**
> > 1. Jointly generating explicit progress-awareness content, reasoning content, and function calls in one response.
> > 2. Generating explicit progress-awareness content in the first stage, followed by reasoning content and function calls in the second stage, while masking out the loss on the progress-awareness content.
> > 3. Generating explicit progress-awareness content in the first stage, followed by reasoning content and function calls in the second stage, and including both stages in the loss.
> >
> > Based on validation performance, we finally adopt the strategy of **jointly generating explicit reasoning content and function calls** for MT-GRPO, and **two-step generation with both steps included in the loss** for Progra. We find that these settings effectively leverage the information produced during generation while ensuring that the model accurately captures both progress-awareness and reasoning objectives.
> >
> >
> >
> > We appreciate the reviewer’s feedback and will make the necessary improvements in the revised version.

---

### Note · Authors · 2025-12-30

I have read and agree with the venue's withdrawal policy on behalf of myself and my co-authors.